JCB Journal of Cell Biology

# mTORC1 activity is supported by spatial association with focal adhesions

Yoana Rabanal-Ruiz[1]*, Adam Byron[2]*, Alexander Wirth[3], Ralitsa Madsen[4], Lucia Sedlackova[1], Graeme Hewitt[5], Glyn Nelson[1], Julian Stingele[6,7], Jimi C. Wills[2], Tong Zhang[8], André Zeug[3], Reinhard Fässler[9], Bart Vanhaesebroeck[4], Oliver D.K. Maddocks[8], Evgeni Ponimaskin[3,10], Bernadette Carroll[11], and Viktor I. Korolchuk[1]

The mammalian target of rapamycin complex 1 (mTORC1) integrates mitogenic and stress signals to control growth and metabolism. Activation of mTORC1 by amino acids and growth factors involves recruitment of the complex to the lysosomal membrane and is further supported by lysosome distribution to the cell periphery. Here, we show that translocation of lysosomes toward the cell periphery brings mTORC1 into proximity with focal adhesions (FAs). We demonstrate that FAs constitute discrete plasma membrane hubs mediating growth factor signaling and amino acid input into the cell. FAs, as well as the translocation of lysosome-bound mTORC1 to their vicinity, contribute to both peripheral and intracellular mTORC1 activity. Conversely, lysosomal distribution to the cell periphery is dispensable for the activation of mTORC1 constitutively targeted to FAs. This study advances our understanding of spatial mTORC1 regulation by demonstrating that the localization of mTORC1 to FAs is both necessary and sufficient for its activation by growth-promoting stimuli.

## Introduction

Growth and proliferation of cells are dependent on two main factors: extracellular stimuli (e.g., growth factors) and building blocks (e.g., free amino acids) to support biosynthetic processes. Mammalian target of rapamycin complex 1 (mTORC1) is the key signaling hub that senses the levels of these growth-promoting cues. The current model postulates that activation of mTORC1 is driven by its translocation from the cytoplasm to the surface of late endosomes/lysosomes (Rabanal-Ruiz and Korolchuk, 2018). This association with endomembranes allows mTORC1 to sense amino acids transported from the extracellular environment as well as those derived by the degradation of proteins within lysosomes. Significant advances have been made in the last decade to understand the molecular mechanisms controlling the recruitment, retention, and release of mTORC1 to and from the lysosomal membrane (Wolfson and Sabatini, 2017).

An additional, yet much less understood, layer of mTORC1 regulation involves the dynamic localization of lysosomes within the cell (Ballabio and Bonifacino, 2020). It has previously been shown that distribution of lysosomes to the cell periphery

facilitates activation of mTORC1, potentially by bringing lysosome-bound mTORC1 into close proximity with nutrient inputs into the cell (Korolchuk et al., 2011; Pu et al., 2016; Rabanal-Ruiz and Korolchuk, 2018). However, the molecular mechanism by which peripheral translocation of mTORC1 stimulates its activity remains unknown. Here, we addressed this question by investigating the functional interactions of mTORC1 within peripheral regions of the cell. We demonstrate that focal adhesions (FAs) represent discrete "growth" signaling hubs, with localization of mTORC1 to FAs being both necessary and sufficient for its activation downstream of growth-promoting stimuli.

## Results

### Proximity labeling identifies association between mTORC1 and FA proteins

To identify spatial associations of mTORC1 in nutrient-replete conditions (in which lysosomes, together with associated mTORC1,

........................................................................................................................................................................

[1]Biosciences Institute, Faculty of Medical Sciences, Newcastle University, Newcastle upon Tyne, UK; [2]Cancer Research UK Edinburgh Centre, Institute of Genetics and Molecular Medicine, University of Edinburgh, Edinburgh, UK; [3]Cellular Neurophysiology, Hannover Medical School, Hannover, Germany; [4]UCL Cancer Institute, University College London, London, UK; [5]DSB Repair Metabolism Laboratory, The Francis Crick Institute, London, UK; [6]Gene Center, Ludwig Maximilians University Munich, Munich, Germany; [7]Department of Biochemistry, Ludwig Maximilians University Munich, Munich, Germany; [8]Wolfson Wohl Cancer Research Centre, Institute of Cancer Sciences, University of Glasgow, Glasgow, UK; [9]Department of Molecular Medicine, Max Planck Institute of Biochemistry, Martinsried, Germany; [10]Institute of Neuroscience, Lobachevsky State University of Nizhni Novgorod, Nizhny Novgorod, Russia; [11]School of Biochemistry, University of Bristol, Bristol, UK.

*Y. Rabanal-Ruiz and A. Byron contributed equally to this paper; Correspondence to Bernadette Carroll: bernadette.carroll@bristol.ac.uk; Viktor I. Korolchuk: viktor.korolchuk@ncl.ac.uk; Y. Rabanal-Ruiz's present address is Department of Medical Sciences, Faculty of Medicine, University of Castilla-la Mancha, Ciudad Real, Spain; Lucia Sedlackova's present address is Centre for Genomic Regulation, The Barcelona Institute of Science and Technology, Barcelona, Spain.



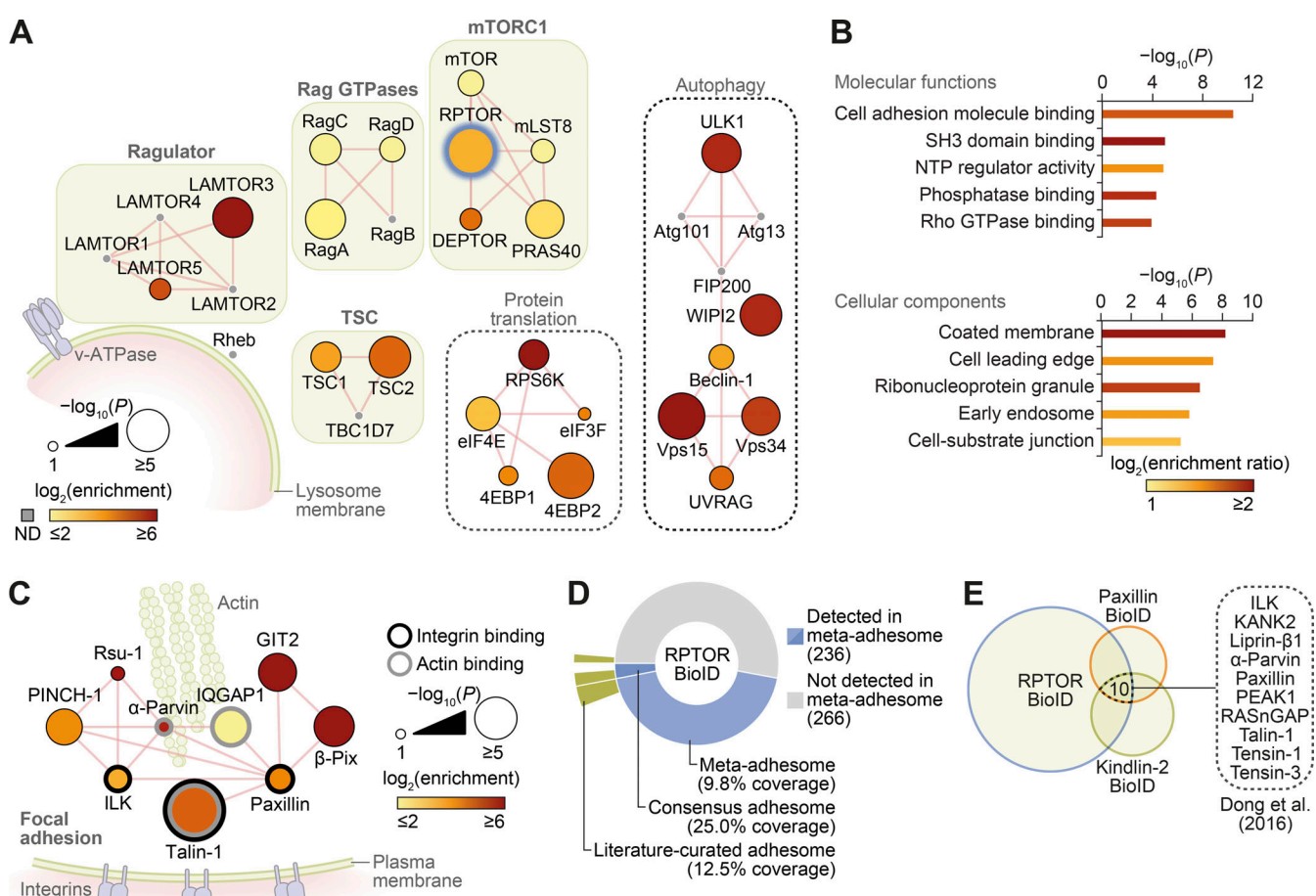

Figure 1. **RPTOR BioID2 reveals spatial association between mTORC1 and FAs. (A)** Network representation of RPTOR BioID2 dataset. Proteins identified include well-known regulators of mTORC1 (protein complexes shown in green boxes), autophagy, and protein translation (black dashed boxes; see Table S1 for a full list). ND, not detected in the proteomics dataset. Thick blue node border indicates BioID2-RPTOR. Edges (red lines) indicate reported physical protein–protein interactions. **(B)** Gene Ontology overrepresentation enrichment analyses of proteins identified by RPTOR BioID2. Molecular functions enriched with P < 10$^{-3}$ and cellular components enriched with P < 10$^{-5}$ are shown (hypergeometric tests with Benjamini–Hochberg correction; 5% FDR threshold). Bar color represents enrichment ratio of overrepresented terms. NTP, nucleoside triphosphatase. **(C)** Network representation of consensus adhesome proteins identified in the RPTOR BioID2 dataset. Edges (red lines) indicate reported physical protein–protein interactions. The largest interconnected subnetwork is shown. **(D)** Proportion of RPTOR-proximal proteins in the meta-adhesome database, including the consensus adhesome (Horton et al., 2015; number of identified proteins indicated in parentheses). Segments are labeled with respective coverage of the meta-adhesome, consensus adhesome, and literature-curated adhesome (Winograd-Katz et al., 2014) by RPTOR-proximal proteins. See also Table S1. **(E)** Identification of 10 proteins significantly enriched in the RPTOR BioID2 dataset that were also identified in published BioID datasets of the FA proteins paxillin and kindlin-2 in U2OS cells (Dong et al., 2016).

are translocated to the cell periphery; Hong et al., 2017; Korolchuk et al., 2011), we employed a proximity labeling and proteomics approach (Kim et al., 2016). Using cells expressing an inducible biotin ligase–fused RPTOR (regulatory associated protein of mTOR Complex 1, or Raptor; BioID2-RPTOR), we performed biotin labeling of proteins proximal to RPTOR in the presence of amino acids and growth factors. Mass spectrometry (MS) analyses of the subsequent streptavidin pull-downs confirmed the presence of core mTORC1 proteins (mTOR, mLST8, and DEPTOR), upstream regulators (Rag GTPases and TSC1/2), as well as downstream substrates (ULK1, 4EBP1, and RPS6K; Fig. 1 A; Fig. S1, A–E; and Table S1). As expected, the most significantly enriched pathway in the proteomics dataset was mTOR signaling, and biological processes related to autophagy and TOR signaling, as well as vesicle transport and GTPase regulation, were significantly overrepresented (Fig. S1, F and G). Interestingly, the most significantly overrepresented molecular function

in the proteomics dataset was cell adhesion molecule binding, in addition to significant enrichment of cellular components related to cell leading edge and cell-substrate junctions (Fig. 1 B). Several proteins involved in FA regulation were particularly prominent (Fig. 1 C; Fig. S1 H; and Table S1).

FAs are large, dynamic protein complexes composed of integrins and a diverse array of scaffolding and signaling molecules that mediate the association of the actin cytoskeleton with the extracellular matrix (Byron and Frame, 2016). Functionally, FAs are involved in cell polarization, spreading, and migration. The dynamic control of FA formation, maturation, disassembly, and recycling is required for development and wound healing and is intimately implicated in cancer cell metastasis (Byron and Frame, 2016). As the BioID technique generates a labeling radius of ~10 nm from the BirA*-tagged bait (based on studies of nuclear pore complex proteins; Kim et al., 2014), our data suggest

that, in nutrient-replete conditions, mTORC1 comes into close proximity of functional FA complexes containing core regulators such as talin-1 (TLN1), paxillin, and integrin-linked kinase, as well as other scaffolding proteins and regulators of actin dynamics such as GIT2, β-PIX (ARHGEF7), and VASP (Fig. 1 C and Table S1; Giannone et al., 2003; Wang et al., 2011; Zaidel-Bar et al., 2003).

Almost half of identified RPTOR BioID2 hits (47%) were present in the "meta-adhesome," a comprehensive, proteomics-based database of adhesion proteins (Horton et al., 2015), including the core FA proteins mentioned above (Fig. 1 D and Fig. S1 H). Furthermore, one quarter of the consensus adhesome of most frequently identified adhesion proteins was detected by RPTOR BioID2 (Fig. 1 D). Conversely, gene set enrichment analysis of the meta-adhesome revealed a significant enrichment of mTORC1-related proteins (P = 4 × 10$^{-6}$; Fig. S1, I and J; and Table S2) and lysosome-related proteins (P = 9 × 10$^{-5}$; Fig. S1, I and K; and Table S3), including mLST8, Rheb, S6, and LAMP1. Interaction network analysis revealed that the majority of mTOR signaling pathway components detected in adhesion complexes formed an interconnected subnetwork, which clustered based on upstream and downstream signaling components (Fig. S1 J). Additional analyses of published BioID-based proteomics datasets for FA proteins paxillin and kindlin-2 identified that 44% and 37%, respectively, of enriched proteins reported by Dong et al. (2016) were also present in the RPTOR BioID2 dataset. 10 proteins were found significantly enriched in all three datasets, including the core FA regulators TLN1 and integrin-linked kinase (Fig. 1 E and Table S1; Dong et al., 2016). A similar overlap of adhesion proteins was also observed in the paxillin and kindlin-2 BioID datasets reported by Chastney et al. (2020; Fig. S1 L and Table S1). Together, these data indicate that there is a significant and previously unrecognized spatial association between mTORC1 and FAs.

### mTORC1 is activated in FAs

Since FAs are plasma membrane–anchored complexes, we hypothesized that the nutrient-dependent dispersal of mTOR-positive lysosomes to the cell periphery (Fig. S2 A) facilitates the mTORC1–FA spatial association. Indeed, we observed increased colocalization of mTOR and LAMP1 with the core FA protein paxillin following starvation–refeeding compared with starvation alone (Fig. 2, A and B), and this correlated with nutrient-dependent changes in lysosome localization (Fig. S2 A). To further validate the nutrient-dependent association between mTOR and paxillin, we used an in situ proximity ligation assay (PLA). We detected proximity between paxillin and mTOR in starved cells, which was increased following refeeding (Fig. 2 C). We therefore conclude that a fraction of mTORC1 exists in close proximity to FAs, which is further enhanced by nutrients, potentially as a result of lysosomal translocation to the cell periphery.

Next, we investigated the spatial distribution of mTORC1 activity. Using a phospho-specific antibody against the mTORC1 substrate S6 (Ser235/236), we detected a significant increase in mTORC1 activity at the edge of the cell in response to either amino acid or growth factor refeeding, which again correlated

with peripheral lysosomal localization (Fig. S2, B and C). At the population level, >40% of cells displayed activation of mTORC1 at the periphery following starvation and refeeding protocols (Fig. S2 D). Importantly, nearly all cells with peripheral lysosomal localization also exhibited peripheral phospho-S6 staining, thus indicating direct correlation between the two events (Fig. S2 E). Furthermore, overexpression of ARL8B, which forces lysosomal translocation toward the plasma membrane, also increased p-S6 levels in the cell periphery as well as the overall proportion of cells with peripheral p-S6 (Fig. S2 F). Peripheral activation of mTORC1 was also accompanied by increased p-S6 in the cytoplasm ("intracellular") and increased total mTORC1 activity as monitored by immunoblotting (Fig. S2, B, C, F and G).

Interestingly, activity of mTORC1 was not uniform along the plasma membrane but was instead concentrated around cell protrusions, which are enriched in FAs (Fig. S2, B, C, and F). Indeed, nutrient refeeding led to a significant and specific increase in colocalization between p-S6 and paxillin at the cell periphery (Fig. 3, A–C; and Fig. S3 A). Our image analyses suggested the existence of two spatially distinct p-S6 pools, intracellular and peripheral, with the latter strongly overlapping with FAs (Fig. 3 C and Fig. S3 A). Further image quantification indicated that, after refeeding with FCS, there is a clear, localized increase in mTORC1 activity in the vicinity of FAs when compared with adjacent areas (Fig. 3, D and E). The observation of this phenotype in different cell types (Fig. 3 A and Fig. S3 B) suggests that peripheral activation of mTORC1 is a widespread process.

To quantify the dynamics of mTORC1 activation, we used the TORCAR (mTORC1 activity reporter) biosensor, a fluorescence resonance energy transfer (FRET)–based reporter of 4EBP1 phosphorylation (Zhou et al., 2015; Fig. 3 F). Using this assay, we confirmed that there is a specific increase in mTORC1 activity in GFP-paxillin–positive regions following refeeding (Fig. 3, G and H; Fig. S3, C and D; and Video 1). Strikingly, mTORC1 activation occurred at a faster rate and higher amplitude in FAs compared with adjacent control areas (Fig. 3, G and H; and Video 1). At the same time, preferential activation of mTORC1 in FA areas was not detected when stimulated with a membrane-permeable leucine analogue as a control (Fig. 3 I). Finally, since mTORC1 activation stimulates protein translation, we used a Click-IT based assay for the detection of newly synthesized proteins (via incorporation of the amino acid analogue L-HPG [L-homoproparglyglycine]) as a functional readout for mTORC1 activity. The spatial distribution of the L-HPG signal in response to refeeding was found to be highly similar to that of p-S6, suggesting that mTORC1 activation in the vicinity of FAs promotes localized protein translation (Fig. 3, J–M). Importantly, specificity of all three readouts of mTORC1 activity (p-S6, TORCAR, and L-HPG) was confirmed using rapamycin (Fig. 3, A, B, E, J, K, and M; and Fig. S3, A and D).

### FAs are sites of growth factor receptor activation

We hypothesized that FAs support localized activation of mTORC1 via a signal transduction mechanism and that FAs may represent previously underappreciated hubs for the input of growth-promoting stimuli into the cell. The observation that no

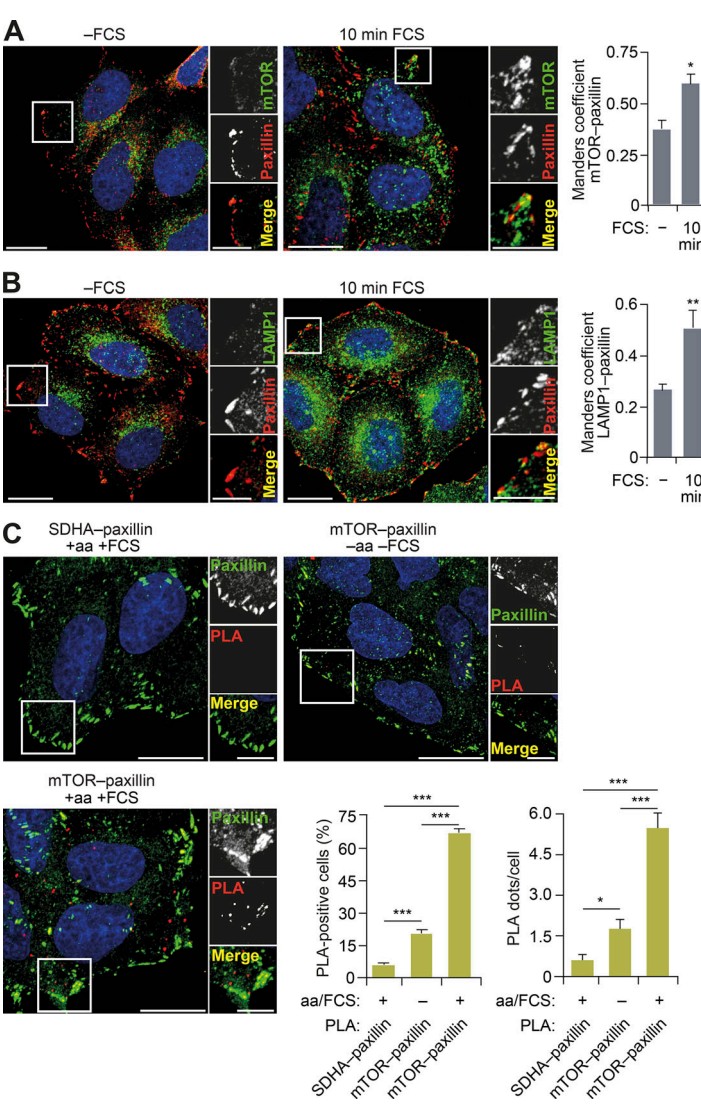

Figure 2. **mTORC1 is localized at FAs. (A and B)** HeLa cells grown in full-nutrient medium were FCS starved for 18 h (−FCS) or FCS starved and then recovered in FCS-containing medium for 10 min (10 min FCS) and immunostained for mTOR (A) or LAMP1 (B) and the FA protein paxillin. Colocalization (Manders coefficient) was analyzed. **(C)** PLAs for mTOR–paxillin interactions (and SDHA–paxillin as a control) were performed in HeLa cells under starvation–refeeding conditions as in A and B. Cells were counterstained with paxillin antibody to mark FAs, and the proportions of PLA-positive cells and PLA dots/cell were quantified. Error bars represent SEM; n = 3 independent experiments. *, P < 0.05; **, P < 0.01; ***, P < 0.001; two-sided Student's t test. Scale bars, 20 µm (insets, 10 µm). Nuclei were visualized with DAPI.

distinct spatial activation of mTORC1 was detected in FAs by the TORCAR FRET biosensor when cells were refed a cell-permeable leucine analogue is in agreement with this hypothesis (Fig. 3 I). Further support comes from the observation that both IGFR1 (insulin-like growth factor 1 receptor) and EGFR (EGF receptor) were specifically activated within paxillin-positive FAs and in isolated FAs (remaining after the removal of cell bodies) in response to refeeding (Fig. 4, A and B; and Fig. S4, A and B). The amino acid transporter SLC3A2 was also found to be tightly localized to FAs (Fig. 4 C and Fig. S4 B). Due to limitations of reagents, we were unable to simultaneously probe for endogenous activated growth factor receptors and mTOR, but using LAMP1 (as a proxy for mTORC1-positive lysosomes, based on Fig. S2 A; Korolchuk et al., 2011), lysosomes were seen to colocalize with both activated growth factor receptors and the amino acid transporter (Fig. S4 C).

In line with these observations, our interaction network analysis revealed that growth factor receptors found in the meta-adhesome were linked to an mTOR signaling subnetwork (Fig. S4 D and Table S2). This subnetwork also contained signaling adaptors and transducers, including PIK3CA (the catalytic p110α subunit of class IA phosphoinositide 3-kinase, or PI3K) and MAPK1 (extracellular signal-regulated kinase 2), which are well-known activators of mTORC1 (Saxton and Sabatini, 2017). Furthermore, in silico analysis suggested associations between adhesion complex–localized growth factor receptor–related proteins with lysosome-associated proteins (Fig. S4 E and Table S3). Based on these observations, we propose an extension to the spatial model of mTORC1 activation to include FAs. Specifically, these data support a novel role for FAs as (1) hubs for mitogenic inputs and (2) the specific location in which mTORC1-positive lysosomes reside. Therefore, FAs are tightly coupled with signal transduction cascades mediating cellular growth factor and nutrient sensing.

## FAs are required for mTORC1 signaling

Next, we tested whether FAs are required for growth factor and nutrient sensing. First, we used a combination of functional, genetic, and pharmacological approaches to disrupt FAs and monitor signaling responses to feeding. In the absence of TLN1 and TLN2, functional FAs cannot form, and in TLN1 and TLN2 double-knockout (DKO) fibroblasts, the addition of amino

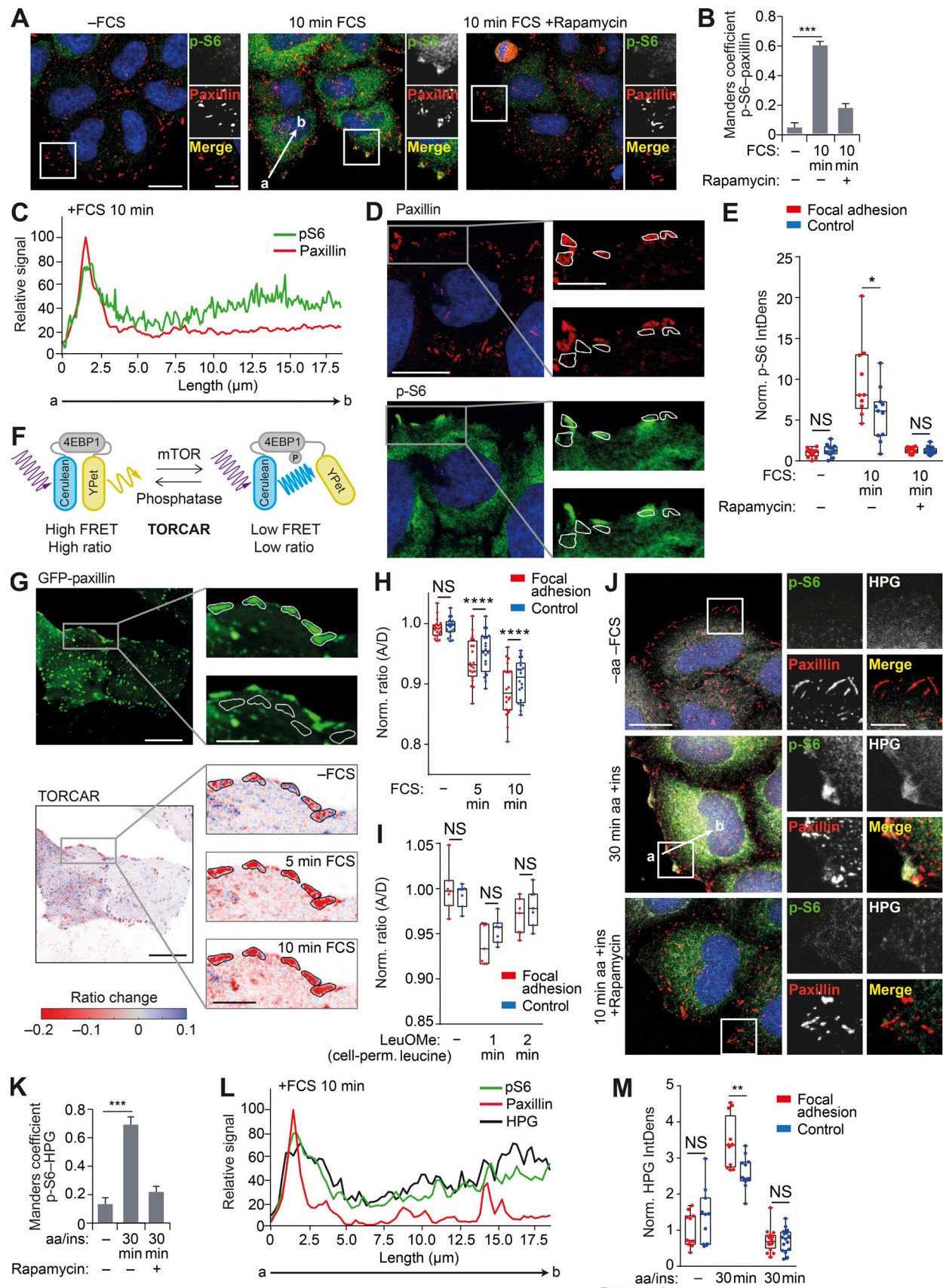

Figure 3.  **mTORC1 is activated in FAs. (A)** HeLa cells grown in full-nutrient medium were FCS starved for 18 h (−FCS) or FCS starved and then recovered in FCS-containing medium for 10 min (10 min FCS) in the absence or presence of the mTORC1 inhibitor rapamycin (100 nM) and immunostained for p-S6 and the

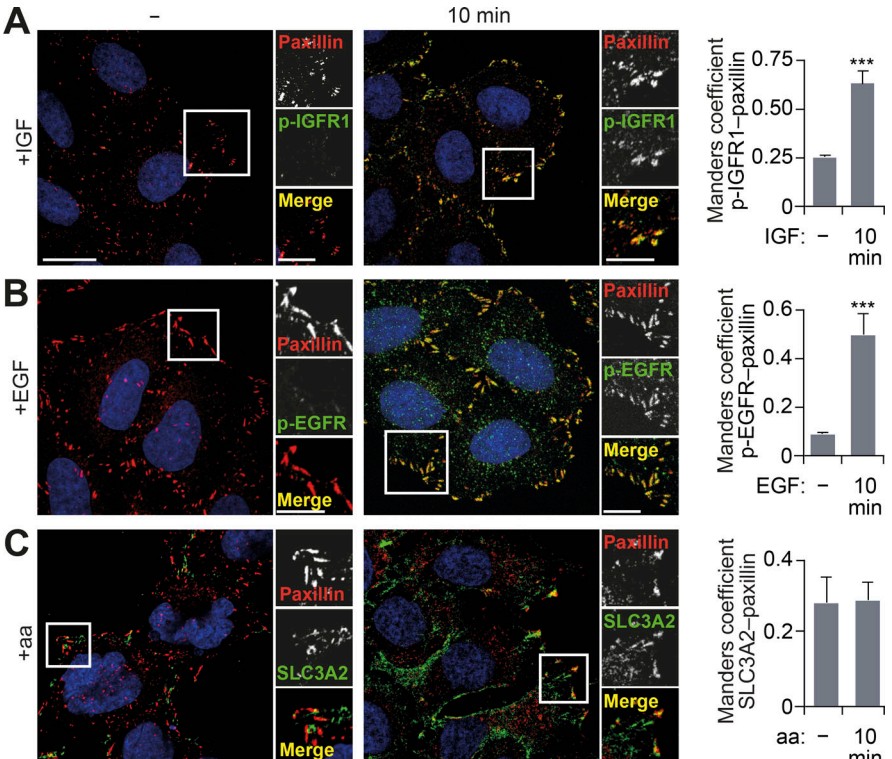

FA protein paxillin. **(B)** Colocalization (Manders coefficient) between p-S6 and paxillin was analyzed. **(C)** Fluorescence intensity line profile plots corresponding to lines exemplified by an arrow in A. **(D and E)** Analysis of p-S6 in ROIs corresponding to FAs and adjacent (control) areas. Representative confocal image of paxillin (top) and p-S6 (bottom) in refed HeLa cells (D). The ROIs corresponding to FAs (top inset) and adjacent (control) areas (bottom inset) are indicated by white borders in zoom insets (D; right), and p-S6 IntDens in ROIs was quantified (E). **(F)** Diagram demonstrating the principle of the TORCAR biosensor (Zhou et al., 2015). **(G)** TORCAR biosensor analysis. Representative confocal image of GFP-paxillin (top) and GFP-paxillin intensity-scaled ratio image of TORCAR FRET-based biosensor (bottom) in serum-starved and refed HeLa cells. Note that to visualize the differences, the images were gamma adjusted, and white borders in GFP-paxillin zoom insets highlight the ROIs corresponding to FAs (top inset) and control areas (bottom inset) used for quantification. **(H and I)** mTORC1 activity was quantified in serum-starved versus refed conditions (H) and before and after stimulation with 25 mM membrane-permeable leucine methylester (I), presented as the normalized ratio of TORCAR biosensor in FAs and control areas. Each data point represents a coverslip including several cells. A/D, acceptor-to-donor ratio. **(J)** HeLa cells grown in full-nutrient medium were starved of amino acids and FCS (−aa −FCS) and then recovered in amino acid– and insulin-containing medium for 30 min (30 min aa +ins) in the presence of L-HPG with and without rapamycin. HPG incorporation was visualized via Click-IT reaction with Alexa Fluor 488 azide before cells were immunostained for p-S6 and paxillin. **(K)** Colocalization (Manders coefficient) between p-S6 and HPG was analyzed. **(L)** Fluorescence intensity line profile plots corresponding to lines as exemplified by an arrow in J. **(M)** Quantification of HPG IntDens in ROIs corresponding to FAs and adjacent (control) areas. Scale bars, 20 μm (insets 10 μm). Nuclei were visualized with DAPI. Error bars represent SEM; $n = 3$ independent experiments. For B, E, K and M, *, $P < 0.05$; **, $P < 0.01$; ***, $P < 0.001$; two-sided Student's $t$ test. For H and I, ****, $P < 0.0001$; two-way ANOVA with Sidak's multiple comparisons test.

acids, with or without FCS, was unable to significantly activate mTORC1 (Fig. 5, A–C). Growth factor signaling upstream of mTORC1 was also suppressed (Fig. 5 C). Similarly, the uptake of amino acids, including the potent mTORC1 activator leucine, was significantly reduced in TLN1/2 DKO cells compared with controls during refeeding (Fig. 5 D). Together, these data indicate an important role for functional FAs in controlling growth factor and amino acid input into the cell as well as downstream signaling response by mTORC1. Similar defects were observed when FAs were disrupted by siRNA-mediated knockdown of TLN1/2 or when cells were treated in suspension (thus without FAs) versus adherent culture (Fig. S5, A–C). Furthermore, pharmacological disruption of FAs with the integrin antagonist cilengitide or a Rho-associated protein kinase inhibitor (ROCKi; Y-27632) also suppressed mTORC1 activation by amino acids and

growth factors (Fig. 5 E; and Fig. S5, D and E). The advantage of these acute pharmacological disruptions of FAs is that they cause less drastic effects on cell morphology, which permitted observation of clear loss of p-S6 at the cell periphery, accompanied by a loss of intracellular p-S6 (Fig. 5 E).

Second, if cells with fewer FAs have reduced mTORC1 activity, we reasoned that the opposite may also be true and that cells with more FAs would support increased mTORC1 activity. We have previously reported that constitutive activation of growth factor and mTORC1 signaling (Carroll et al., 2017) is a defining feature of cellular senescence, a potent tumor suppressor mechanism that is further characterized by irreversible exit from the cell cycle and increased secretion of inflammatory factors (Carroll and Korolchuk, 2018). Here, we show for the first time that senescent cells contain significantly more

Figure 4. **Growth factor signaling is concentrated at FAs. (A–C)** HeLa cells grown in full-nutrient medium were serum starved (for p-EGFR and p-IGFR1 staining) or amino acid starved (for SLC3A2 staining) and then stimulated as indicated. Cells were fixed and immunostained for p-IGFR1 (A), p-EGFR (B), or SLC3A2 (C) and paxillin, and colocalization was quantified. Error bars represent SEM; $n = 3$ independent experiments. Scale bars, 20 μm (insets, 10 μm). Nuclei were visualized with DAPI. ***, $P < 0.001$; two-sided Student's $t$ test.

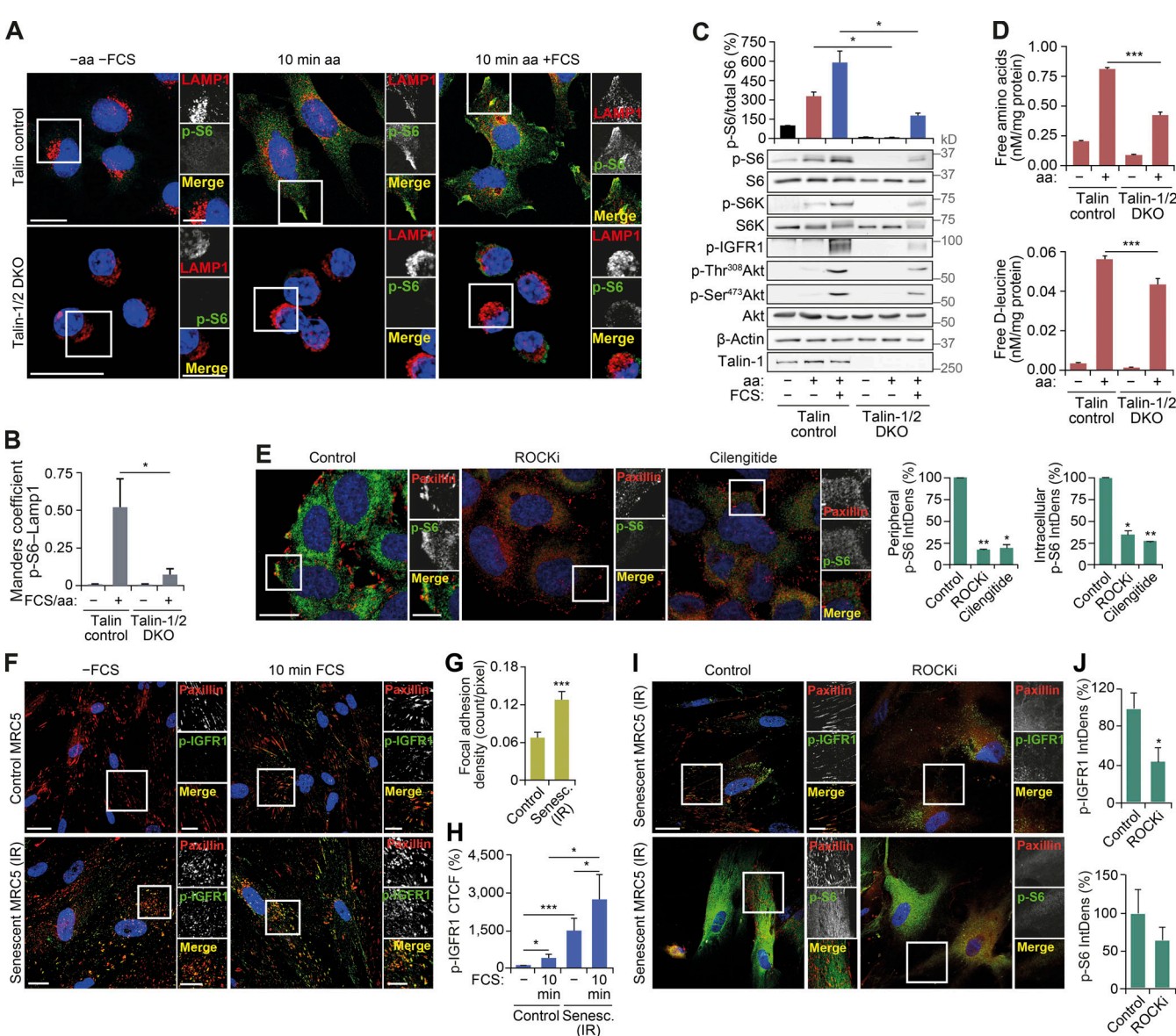

Figure 5. **Functional FAs are required for mTORC1 activation. (A–C)** Control and TLN1/2 DKO mouse kidney fibroblasts grown in full-nutrient medium were amino acid and FCS starved (−aa −FCS) and then recovered in amino acid–containing medium (10 min aa) or amino acid– and FCS-containing medium (10 min aa +FCS) for 10 min. Cells were analyzed by immunostaining for LAMP1 and p-S6 (A), colocalization (Manders coefficient) was quantified (B), and cell lysates were analyzed by immunoblotting to monitor changes in the mTORC1 pathway (C). **(D)** Control and TLN1/2 DKO cells subjected to the starvation and refeeding as in A–C were lysed and analyzed by liquid chromatography MS to measure intracellular levels of amino acids. Amino acid concentrations were normalized to protein levels. **(E)** HeLa cells were FCS starved for 18 h (−FCS), treated with 50 µM ROCKi or 5 µM integrin antagonist (cilengitide) for 1 h in −FCS medium, starved of amino acids (−aa −FCS) for 1 h in the presence of inhibitors, and then recovered in full-nutrient medium for 10 min. Cells were subjected to immunostaining for paxillin and p-S6, and the IntDens of peripheral or intracellular p-S6 staining was quantified. **(F–H)** Control and senescent (30 d after 20-Gy x-ray irradiation [IR]) primary human fibroblasts were subjected to FCS starvation for 18 h (−FCS) or FCS starved and then recovered in FCS-containing medium for 10 min (10 min FCS), fixed and immunostained for p-IGFR1 and paxillin (F). Number of FAs per pixel (G) and CTCF of p-IGFR1 staining (H) were quantified. Note that CTCF was used because it takes into account cell size as senescent cells are significantly larger than proliferating fibroblasts. **(I and J)** Senescent (30 d after 20-Gy x-ray irradiation) primary human fibroblasts were subjected to FCS starvation for 18 h (−FCS); during last 2 h of starvation, cells were treated with DMSO (control) or 50 µM ROCKi. Cells were subjected to immunostaining for paxillin and p-IGFR1 or p-S6 (I), and the IntDens of p-IGFR1 and p-S6 staining was quantified (J). Error bars represent SEM; n = 3 independent experiments. *, P < 0.05; **, P < 0.01; ***, P < 0.001; two-sided Student's t test. Scale bars, 20 µm (insets, 10 µm). Nuclei were visualized with DAPI.

paxillin-positive FAs compared with proliferative cells (Fig. 5, F and G). This is accompanied by higher levels of phosphorylated IGFR1 during starvation (Fig. 5 H). The persistent activity of IGFR1 was localized to FAs and was significantly reduced upon treatment with ROCKi, which also suppressed constitutive

mTORC1 activity in senescent cells (Fig. 5, I and J). The partial effect of ROCKi on S6 phosphorylation is consistent with our previous study demonstrating that the constitutive mTORC1 activity in senescent cells is only partially dependent on growth factor signaling but is also a result of increased intracellular

concentrations of amino acids compared with proliferating cells (Carroll et al., 2017). While we previously demonstrated a role for lysosomal degradation in generating these amino acids, our current observations that FAs affect intracellular amino acid concentrations, and the presence of more FAs in senescent cells, together suggest that increased amino acid uptake may also contribute to the higher levels seen in senescence (Carroll et al., 2017).

**mTORC1 is activated by lysosomal positioning in proximity to FAs**

Disruption of FAs suppressed growth factor signaling and amino acid uptake upstream of mTORC1 (Fig. 5, A–D) but can also have other pleiotropic effects on cellular physiology, such as cytoskeletal changes (Byron and Frame, 2016). Therefore, we investigated if uncoupling of stimulus-dependent mTORC1 recruitment to FAs would suppress its activity without affecting upstream signaling. As lysosomes transport mTORC1 to FAs in response to feeding (Fig. 2), we performed knockdown of ARL8B, which prevented peripheral distribution of mTORC1-positive lysosomes in response to feeding and, consistent with previous studies, was found to inhibit mTORC1 activation (Fig. S5 F; Jia and Bonifacino, 2019; Korolchuk et al., 2011). To test if forced localization of mTORC1 to FAs is sufficient to rescue this phenotype, we generated RPTOR constructs fused to FLAG-tag alone or FLAG and the FA targeting (FAT) sequence from vinculin (Fig. 6 A). When expressed in HeLa cells, both constructs could form a complex with mTOR (Fig. S5 G). Consistent with the function of the FAT domain, RPTOR[FAT], but not the control RPTOR construct, strongly colocalized with paxillin without affecting its distribution and displayed increased interaction with paxillin by immunoprecipitation assays (Fig. 6 A; and Fig. S5, G and H).

In cells expressing RPTOR, ARL8B depletion significantly decreased peripheral and intracellular mTORC1 activation as monitored by p-S6 immunofluorescence and immunoblotting (Fig. 6, B–D), suggesting that mTORC1 activation at the cell periphery may, at least in part, contribute to intracellular activation of mTORC1 and support overall mTORC1 activity. Consistent with previous reports (Jia and Bonifacino, 2019), depletion of ARL8B also suppressed mTORC2 activity; however, it did not affect FA distribution within the cell or growth factor signaling upstream of mTORC1 (Fig. 6 D; and Fig. S5, I and J). In contrast, knockdown of ARL8B did not suppress mTORC1 activity in cells expressing RPTOR[FAT] (Fig. 6, B–D). This effect was seen both at short (10 min) and long (60 min) refeeding times (Fig. 6, B–D; and Fig. S5 J). Therefore, targeting mTORC1 to FAs is sufficient for its activation by nutrients regardless of its localization to lysosomes. Taken together, our data strongly indicate that FAs are required both for proper growth factor signaling/amino acid uptake and for efficient downstream activation of mTORC1.

## Discussion

Our current study conceptually extends the existing model of growth factor and nutrient signaling. We demonstrate that growth factor receptors are specifically activated within FAs

(Fig. 6 E). While previous reports have demonstrated the importance of integrin-based FAs for proper growth factor signaling (Eberwein et al., 2015; Ivaska and Heino, 2011; Yamada and Even-Ram, 2002), here, we have shown that activation of IGFR1 and EGFR is spatially restricted to FAs. Furthermore, at least some amino acid transporters are enriched in FAs, making them an important gateway for the uptake of key nutrients into the cell. This provides a biological rationale for our finding that mTORC1 is activated specifically in FA-enriched regions and that FAs are necessary for the full activation of mTORC1. It also provides an explanation for the positive effect of nutrient-dependent peripheral lysosomal distribution on mTORC1 activity (Korolchuk et al., 2011). Interestingly, forcing mTORC1 to FAs allows for its full activation by nutrients in the absence of lysosomes at the cell periphery. This suggests that coupling of mTORC1 with growth factor signaling and amino acid input into the cell may be sufficient even when mTORC1 is not on lysosomes.

Our findings are consistent with the growing evidence that mTORC1 can be activated and functions in spatially distinct pools. For example, in yeast, TORC1 present on vacuoles (equivalent to lysosomes) regulates protein synthesis, while endosomally localized TORC1 regulates autophagy processes (Hatakeyama and De Virgilio, 2019). Furthermore, plasma membrane–targeted mTORC1 has previously been shown to be active (Sancak et al., 2010). It remains to be investigated how activation of mTORC1 in the vicinity of FAs is governed by the currently known factors such as Rag and Rheb GTPases and whether it involves new molecular players. However, our data strongly suggest that this process is important for the overall cellular sensing of growth-promoting stimuli, warranting further investigation into the molecular mechanisms of mTORC1 activation in FAs (and potentially other related structures such as fibrillar adhesions; Rainero et al., 2015) and its relationship with mTORC1 signaling on endomembranes.

It also remains to be seen whether activation of mTORC1 at FAs drives a specific, localized function, but the identification of protein translation regulators, such as S6, eIF4B, and eIF4E, and autophagy initiation regulators, such as ULK1, in proximity to FAs suggests that this is a possibility. This is further supported by our observation that mTORC1 drives localized protein translation in the FA-enriched areas of the cell. It has previously been shown that inactivation of the total mTORC1 pool by inhibitors such as rapamycin (Liu and Parent, 2011) and perturbed transport of lysosomes to the cell periphery (Schiefermeier et al., 2014; Guardia et al., 2016) interfere with FA dynamics and cell migration. Similarly, suppression of mTORC1 by nutrient deprivation increases internalization of adhesion complexes and their transport to spatially localized lysosomes (Rainero et al., 2015). Future work will establish whether the physical presence of mTORC1 proximal to FAs contributes to these phenotypes, for example by activating local synthesis of proteins required for adhesion and migration. Conversely, previous reports have identified roles for autophagy in regulating the dynamics of FAs (Kawano et al., 2017; Vlahakis and Debnath, 2017; Xu and Klionsky, 2016); for example, detachment of cells from the extracellular matrix activates selective autophagic degradation of

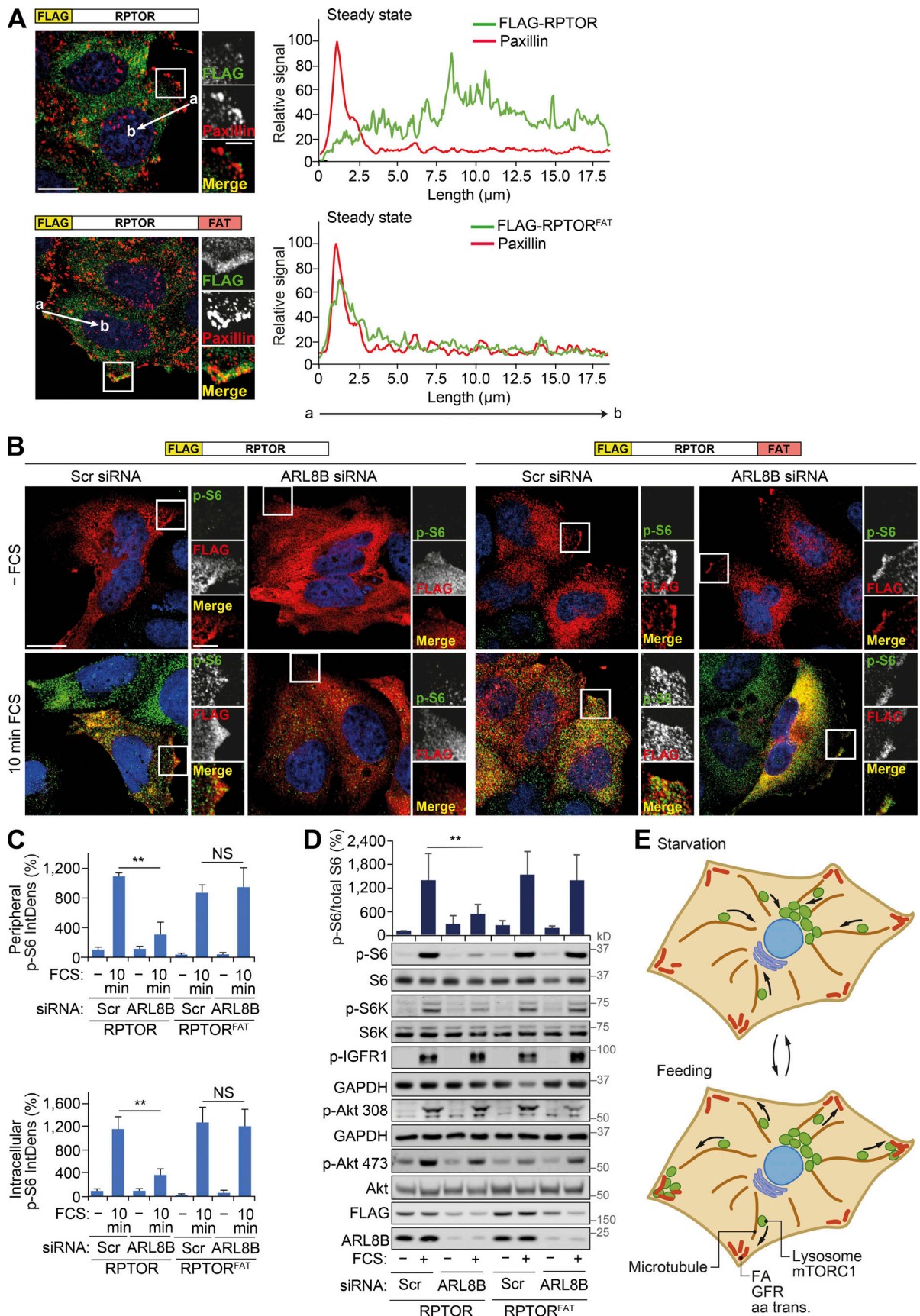

Figure 6. **Constitutive targeting of mTORC1 to FAs uncouples it from regulation by lysosomal positioning. (A)** HeLa cells expressing FLAG-RPTOR or FLAG-RPTOR[FAT] grown in full-nutrient medium were immunostained for FLAG and the FA protein paxillin. Representative images and fluorescence intensity

line profile plots corresponding to lines exemplified by arrows are shown. **(B)** Scrambled (Scr) or ARL8B siRNA–transfected HeLa cells expressing FLAG-RPTOR or FLAG-RPTOR[FAT] grown in full-nutrient medium were FCS starved for 18 h (–FCS) or FCS starved and then recovered in FCS-containing medium for 10 min (10 min FCS) and immunostained for p-S6 and FLAG. **(C)** The IntDens of peripheral or intracellular signal of p-S6 was quantified. **(D)** Cells treated as in B were subjected to immunoblot analysis using antibodies as shown. Error bars represent SEM; $n$ = 3 independent experiments (for IntDens, $n \geq 10$ cells were quantified per experiment). **, $P < 0.01$; two-sided Student's $t$ test. Scale bars, 20 μm (insets, 10 μm). Nuclei were visualized with DAPI. **(E)** Diagram illustrating the proposed role of FAs in the activation of mTORC1 in response to growth factor–promoting stimuli. aa trans., amino acid transporter; GFR, growth factor receptor. See Discussion for further details.

FA proteins via the receptor protein NBR1 (Kenific et al., 2016), while detachment-induced ER stress can inhibit mTORC1 and thus activate autophagy via the AMPK–TSC2 axis (Avivar-Valderas et al., 2013). Activation of mTORC1 at FAs by nutrients may therefore suppress autophagy of FA components and thus support cell migration. As such, the work presented here adds a new dimension to the model of reciprocal regulation between mTORC1–autophagy–lysosomes and FAs.

## Materials and methods

### Cell lines
Human HeLa cervical carcinoma cells were cultured in RPMI 1640 medium (R0883; Sigma-Aldrich) or DMEM (D6429; Sigma-Aldrich) with 2 mM L-glutamine, 10% FBS, and 100 U/ml penicillin–streptomycin at 37°C, 5% $CO_2$. HEK293, Cos7, U2OS cell lines and primary human fibroblasts (MRC5) were cultured in DMEM (D6546; Sigma-Aldrich) with 2 mM L-glutamine, 10% FBS, and 100 U/ml penicillin–streptomycin at 37°C, 5% $CO_2$. Tln1[flox/flox] (control) and Tln1[flox/flox] Tln2[−/−] mouse kidney fibroblasts (Theodosiou et al., 2016) were cultured in DMEM–F12 (12-719; Lonza) with 10% FBS and 100 U/ml penicillin–streptomycin at 37°C, 5% $CO_2$.

### siRNA
ON-TARGETplus SMARTpool siRNA against human *ARL8B* (L-020294-01), *TLN1* (L-012949-00), and *TLN2* (L-012909-00) were purchased from Dharmacon. Final siRNA concentrations of 50 or 100 nM were used for silencing, and transfections were performed using Lipofectamine 2000 (Invitrogen) as per company instructions.

### Serum starvation and recovery
Serum starvation was performed overnight (18 h) and recovered by resupplementation of media containing 10% FCS for 10 min. Where indicated, amino acid starvation was performed for 1 h (after one wash with PBS) in RPMI 1640 without amino acids (US Biologicals), and recovery was achieved by addition of 1× RPMI 1640 amino acid solution (R7131; Sigma-Aldrich) with or without 10% FCS, pH 7.2, for 10 min. Where indicated, cells were incubated for the last 2 h before harvest/fixation with the ROCKi Y-27632 (50 μM; SCM075; Sigma-Aldrich) or the integrin-binding peptide cilengitide (5 μM; SML1594; Sigma-Aldrich). Where indicated, cells were preincubated with 100 nM rapamycin for 2 h.

### Immunoblotting
Immunoblotting was performed as described previously (Carroll et al., 2016). In brief, cells were lysed in RIPA buffer (50 mM Tris-HCl, pH 7.4, 150 mM NaCl, 1% NP-40, 0.5% sodium deoxycholate, and 0.1% SDS) supplemented with Halt protease and phosphatase inhibitors (1861280; Thermo Fisher Scientific) on ice. Protein concentrations of lysates were measured using DC protein assay (500-0112; Bio-Rad Laboratories), and equal amounts of protein (20–40 μg) were subjected to SDS-PAGE and immunoblotted. The following primary antibodies were used: rabbit anti–phospho-S6K Thr389 (1:1,000; #9205), rabbit anti-S6K (1:1,000; #9202), rabbit anti–phospho-S6 Ser235/236 (1:2,000; #4856), rabbit anti-S6 (1:2,000; #2217), rabbit anti–phospho-4EBP1 Thr37/46 (1:2,000; #2855), rabbit anti-talin (1:1,000; #4021), rabbit anti-mTOR (1:1,000; #2972), rabbit anti–phospho-Akt Ser473 (1:1,000; #9271), rabbit anti–phospho-Akt Thr308 (1:1,000; #4056), and mouse anti–pan-Akt (1:1,000; #2920), all purchased from Cell Signaling Technology; mouse anti–β-actin (1:10,000; STJ96930), rabbit anti–phospho-EGFR Tyr1016 (1:1,000; STJ91037), rabbit anti-EGFR (1:1,000; STJ96946), and rabbit anti–phospho-IGFR1 Tyr1165/1166 (1:1,000; STJ90299) were purchased from St John's; and additional antibodies were rabbit anti-ARL8B (1:1,000; 13049-1-AP; Proteintech), mouse anti–α-tubulin (1:5,000; 12G10; Developmental Studies Hybridoma Bank), mouse anti-FLAG M2 (1:2,000; F3165; Sigma-Aldrich), and mouse anti-myc (1:1,000; clone 9E10; Upstate). Secondary antibodies conjugated to horseradish peroxidase were all used at 1:5,000 for 1 h at room temperature. Clarity Western ECL substrate (Bio-Rad Laboratories) was used to visualize chemiluminescence on LAS4000 (Fujifilm). Quantification of blots was performed using ImageJ (version 1.41; National Institutes of Health).

### Immunofluorescence
Immunofluorescence was performed essentially as described previously (Carroll et al., 2016). In brief, cells were fixed in 4% formaldehyde in PBS for 10 min, permeabilized with 0.5% Triton X-100 for 10 min, and blocked in 5% normal goat serum/PBS with 0.05% Tween for 1 h, all at room temperature. The following primary antibodies were used: rabbit anti–phospho-S6 Ser235/236 (1:200; 4856; Cell Signaling Technology), rabbit anti-S6 (1:1,000; 2217; Cell Signaling Technology), rabbit anti-mTOR (1:200; 2972; Cell Signaling Technology), rabbit anti-LAMP1 (1:1,000; ab24170; Abcam), mouse anti-LAMP1 (1:1,000; Developmental Studies Hybridoma Bank), mouse anti-paxillin (1:400; 610055; BD Biosciences, which was discontinued during the course of the project), mouse anti-paxillin (5H11; 1:1,000; AHO0492; Sigma-Aldrich), rabbit anti–phospho-EGFR Tyr1016 (1:1,000; STJ91037; St John's), rabbit anti-EGFR (1:1,000; STJ96946; St John's), rabbit anti–phospho-IGFR1 Tyr1165/1166 (1:1,000; STJ90299; St John's), rabbit anti-SLC3A2 (1:200; 47213; Cell Signaling Technology), and rat anti-LAMP1 (for mouse cells,

1:1,000; Developmental Studies Hybridoma Bank). Cells were washed and incubated with the appropriate secondary antibodies (1:1,000; Thermo Fisher Scientific) for 1 h at room temperature. Coverslips were mounted on slides with Prolong Gold antifade reagent with DAPI (Thermo Fisher Scientific) or Fluoroshield mounting medium without DAPI (Abcam). Secondary antibodies were conjugated with Alexa Fluor 488 or 594. Confocal images were collected on an SP8 microscope (Leica Microsystems) using a 63× Plan-Apo/1.4-NA oil objective at room temperature. The acquisition software was Leica LasX. All analyses were performed in ImageJ using regions of interest on z-projected images to determine cell intensity in the respective fluorescence channels. For presentation purposes, stacks were z-projected to maximum intensity, and the brightness was adjusted in ImageJ or Imaris. The same adjustments were made across all images.

## FA isolation
FAs were isolated as previously described (Kuo et al., 2012). Briefly, HeLa cells grown on coverslips were rinsed with PBS and hypotonically shocked for 3 min in triethanolamine-containing low ionic strength buffer (2.5 mM triethanolamine, pH 7.0). Then, cell bodies were removed with hydrodynamic force, and FAs were fixed and immunostained using mouse anti-paxillin (5H11; 1:1,000; AHO0492; Sigma-Aldrich), rabbit anti–phospho-EGFR Tyr1016 (1:1,000; STJ91037; St John's), rabbit anti–phospho-IGFR1 Tyr1165/1166 (1:1,000; STJ90299; St John's), and rabbit anti-SLC3A2 (1:200; 47213; Cell Signaling Technology) antibodies.

## In situ PLA
Proximal associations of mTOR with paxillin were detected using Duolink in situ PLA technology (Sigma-Aldrich) following the manufacturer's instructions. Briefly, fixed and permeabilized HeLa cells were blocked with Duolink blocking solution and incubated with mTOR and paxillin antibodies overnight at 4°C in Duolink antibody diluent. As a negative control, an antibody against mitochondrial protein SDHA was used instead of mTOR. Cells were then washed and incubated with Duolink PLA probes (PLA probe anti-mouse Minus and PLA probe anti-rabbit Plus) in a preheated humidity chamber for 1 h at 37°C. Coverslips were incubated with the ligation solution, washed, and incubated with the amplification solution. Finally, coverslips were incubated with anti-mouse Alexa Fluor 488 secondary antibody to visualize paxillin and mounted on slides with Prolong Gold antifade reagent with DAPI (Thermo Fisher Scientific). PLA signals were detected with a Texas-Red filter and paxillin with a GFP filter on a DM5500 widefield fluorescence microscope (Leica Microsystems) using a 40× objective.

## BioID2-based proximity labeling
InsertTAG-pcDNA FRT/TO (FLP recombination target/TetO) was created by mutagenesis of YFP-pcDNA5 FRT/TO (Ahel et al., 2009) to replace YFP with NheI and XhoI restriction sites using the primers 5′-CCCTCGAGGATATCACAAGTTTGTACAAAAAAGC-3′ (forward) and 5′-TGGCTAGCAAACGCTAGAGTCCGGAG-3′ (reverse) and the Q5 Site-Directed Mutagenesis Kit (E0554S;

New England Biolabs). BioID2-pcDNA5 FRT/TO was created by restriction digestion of BioID2 from myc-BioID2-MCS (74223; Addgene) using NheI and XhoI, followed by ligation into insertTAG-pcDNA FRT/TO. BioID2-RPTOR-pcDNA5 FRT/TO was created by shuttling RPTOR from R77-E301 Hs.RPTOR (70585; Addgene) into BioID2-pcDNA5 FRT/TO using Gateway LR Clonase II enzyme mix (11791100; Thermo Fisher Scientific). BioID2 and BioID2-RPTOR U2OS FlpIn TRex cells were produced by transfecting U2OS FlpIn TRex host cells with BioID2-pcDNA5 FRT/TO or BioID2-RPTOR-pcDNA5 FRT/TO and pOG44 at a ratio of 1 μg pcDNA5 FRT/TO to 3 μg pOG44. Cells were then selected in 250 μg/ml hygromycin (10687010; Thermo Fisher Scientific). Cells were maintained in DMEM with 10% tetracycline-free FBS, 250 μg/ml hygromycin, and 5 μg/ml blasticidin at 37°C, 5% $CO_2$. BioID2-RPTOR expression was induced by incubation with 1 μg/ml tetracycline for a total of 48 h (or DMSO for the control). Proximity labeling was carried as previously described (Kim et al., 2016). Briefly, cells were incubated with 50 μM biotin (in DMSO) in biotin-depleted full-nutrient medium (prepared by prior overnight incubation with NeutrAvidin) for 16 h at 37°C, 5% $CO_2$. Cells were washed twice with cold PBS and lysed in RIPA buffer supplemented with 2× phosphatase and protease inhibitors on ice. Cell lysates were transferred to low-protein-binding microcentrifuge tubes (0030108116; Eppendorf) and incubated with NeutrAvidin agarose beads (29200; Thermo Fisher Scientific) overnight at 4°C. The beads were washed three times with RIPA buffer and two times with PBS.

## MS-based proteomics
Isolated bead-bound proteins from four independent proximity-based labeling experiments (BioID2-RPTOR or control) were incubated with digestion buffer (0.3 μg trypsin in 2 M urea, and 50 mM Tris-HCl, pH 8) for 30 min at 27°C. The supernatant was retained, and beads were washed with digestion buffer supplemented with 10 mM dithiothreitol, combined with the initial supernatant and incubated for 16 h at 37°C in a wet chamber. Iodoacetamide was added to 55 mM (final concentration), incubated for 30 min at room temperature and acidified with trifluoroacetic acid. Peptides were cleaned on StageTips (Rappsilber et al., 2007), eluted with 80% acetonitrile, 0.1% trifluoroacetic acid, and concentrated by vacuum centrifugation. Peptides (one-fifth bead input) were injected into an UltiMate 3000 RSLCnano system coupled to a Q Exactive Plus mass spectrometer (Thermo Fisher Scientific). Chromatographic separation was performed on a home-pulled, home-packed C18 analytical column over a 40-min gradient of 2–40% acetonitrile in 0.5% acetic acid. Eluting peptides were ionized at +1.8 kV before data-dependent analysis on the Q Exactive Plus mass spectrometer. Full scans were acquired with a range of 300–2,000 $m/z$ at a resolution of 70,000, and the top 12 ions were selected for fragmentation with normalized collision energy of 30 and an exclusion window of 10s. Fragment scans were collected at a resolution of 17,500. Automatic gain control target values for full and fragment scans were $3 \times 10^6$ and $10^5$ ions, respectively, and all spectra were acquired with one microscan and without lockmass. The proteomic MS data were deposited to the ProteomeXchange Consortium via the PRIDE partner

repository (Perez-Riverol et al., 2019) with the dataset identifier PXD012795.

## Proteomic data analysis

Label-free quantitative analysis of proteomic MS data were performed using MaxQuant (version 1.6.3.4; Cox and Mann, 2008). Peptide lists were searched against the human UniProtKB database (version 2019_01) and a common contaminants database (supplemented with the chicken avidin sequence) using the Andromeda search engine (Cox et al., 2011). Cysteine carbamidomethylation was set as a fixed modification; methionine oxidation and protein N-terminal acetylation were set as variable modifications (up to five modifications per peptide). Peptide identifications in one or more liquid chromatography–MS runs that were not identified in some runs within a 0.7-min time window were matched and transferred between runs. Peptide and protein false discovery rates (FDRs) were set to 1%, determined by searching a reversed database. Enzyme specificity was set as C terminal to arginine and lysine, except when followed by proline, and up to two missed enzymatic cleavages were allowed in the database search. The minimum peptide length requirement was seven amino acids. At least one peptide ratio was required for label-free quantification, and normalization by the label-free quantification algorithm was skipped. Label-free quantification intensities for proteins quantified in at least two of the four biological replicates were binary-logarithm transformed, from which, for each sample, the transformed intensity of pig trypsin was subtracted from all other proteins in the respective sample. Proteins matching to the reversed or common contaminants databases or only identified by a post-translational modification site were excluded from further analysis. Missing values were imputed sample-wise from a width-compressed, down-shifted normal distribution using Perseus (version 1.5.2.6; Tyanova et al., 2016). Unsupervised agglomerative hierarchical cluster analysis was performed on the basis of Spearman rank correlation computed with a complete-linkage matrix using Cluster 3.0 (C Clustering Library, version 1.54; de Hoon et al., 2004) and visualized using Java TreeView (version 1.1.5r2; Saldanha, 2004). All four independent experiments were well correlated (RPTOR BioID Spearman rank correlation coefficients >0.91; control Spearman rank correlation coefficients >0.81; Fig. S1 C). Specifically enriched proteins were identified using a two-tailed Welch's $t$ test with a permutation-based FDR threshold of 1% (computing 1,000 randomizations) and an artificial within-groups variance threshold of 2. Putative contaminant proteins previously reported in more than one agarose bead–based experiment using U2OS cells in the contaminant repository for affinity purification database (Mellacheruvu et al., 2013; 15% cutoff frequency) were excluded from further analysis.

## Enrichment analyses

Gene names for specifically enriched, noncontaminant proteins identified by MS-based proteomics or all proteins in the meta-adhesome database (Horton et al., 2015) were extracted and used for gene set enrichment analysis (Subramanian et al., 2005). Overlap was computed with curated Kyoto Encyclopedia of Genes and Genomes (KEGG) gene sets in the Molecular Signatures Database (version 6.2; Liberzon et al., 2011). Reported P values were determined using the Kolmogorov–Smirnov statistic and controlled for FDR. Overrepresentation enrichment analysis for Gene Ontology terms was performed using WebGestalt (version 35439c82; Zhang et al., 2005). For analysis of interaction network clusters, weighted set cover redundancy reduction was implemented. Reported P values were determined using the hypergeometric test with Benjamini–Hochberg correction.

## Interaction network analysis

mTOR signaling pathway components annotated in the KEGG knowledgebase (KEGG pathway identifier hsa04150) were parsed and corrected using CyKEGGParser (Nersisyan et al., 2014). Lysosome-associated proteins were extracted from the Gene Ontology database (Gene Ontology cellular component term 0005764). Composite functional association networks based on predicted and physical interactions and pathway associations were generated using GeneMANIA (version 3.4.1; Warde-Farley et al., 2010). Adhesion complex proteins and associated metadata in the meta-adhesome database (Horton et al., 2015) were mapped onto the interaction networks, and mTOR signaling pathway–associated and lysosome-associated adhesion protein subnetworks were extracted and analyzed using Cytoscape (version 3.7.1; Shannon et al., 2003). To interrogate RPTOR-proximal proteins, an interaction network based on physical interactions with specifically enriched, noncontaminant proteins identified by MS-based proteomics was generated using GeneMANIA. Adhesion complex proteins in the literature-curated adhesome (Winograd-Katz et al., 2014), the consensus adhesome (Horton et al., 2015), and the meta-adhesome database were mapped onto the interaction network (Byron, 2018), from which subnetworks were extracted. The interaction network was clustered using the prefuse force-directed algorithm implemented in Cytoscape. To interrogate FA–related growth factor receptors, growth factor receptors were extracted from the Gene Ontology database, filtered for presence in the meta-adhesome database, and used to seed a weighted composite functional association network based on predicted and physical interactions, pathway associations, and reported colocalization using GeneMANIA. These networks were merged with a network of the top 100 highest-scoring proteins associated with all known mTORC1/2 components generated using GeneMANIA and mTOR signaling pathway components annotated in the KEGG knowledgebase or with lysosome-associated proteins extracted from the Gene Ontology database using Cytoscape.

## MS-based metabolomics

HeLa cells were seeded (in triplicate) in 6-well plates and cultured in standard RPMI 1640 until 90% confluent. Cells were treated as indicated. Cells were washed once with cold PBS and lysed (50% methanol, 30% acetonitrile, and 20% deionized $H_2O$) at a concentration of $2 \times 10^6$ cells/ml. Samples were vortexed for 45 s and centrifuged at 13,000 rpm. Cleared lysates were subjected to liquid chromatography–MS as follows, using a

three-point calibration curve with universally labeled carbon-13/nitrogen-15 amino acids for quantification. Cell lysate solution (10 µl) was injected into an Accela 600 LC system coupled to an Exactive mass spectrometer (Thermo Fisher Scientific). Chromatographic separation was performed on a Sequant ZIC-HILIC column (150 × 4.6 mm, 3.5 µm; Merck) with mobile phase A (water) and B (acetonitrile), both with 0.1% formic acid, at a flow rate of 0.3 ml/min. A gradient elution program was used: mobile phase A increasing from 20–80% in 30 min and holding A at 92% for 5 min, followed by 10-min re-equilibration with 20% A. The Exactive mass spectrometer was equipped with a heated electrospray ionization source and operated in an electrospray ionization–positive and –negative switching mode with a scan range of 70–1,200 m/z at a resolution of 50,000. The obtained MS raw data were converted into .mzML files using ProteoWizard (Chambers et al., 2012) and imported into MZMine 2 (version 2.10; Pluskal et al., 2010) to conduct peak extraction, sample alignment, and metabolite identification.

### TORCAR FRET-based biosensor for measuring mTORC1 activity
HeLa cells were cultured and cotransfected with the mTORC1 biosensor TORCAR (pcDNA3-TORCAR; a gift from Jin Zhang, The Johns Hopkins School of Medicine, Baltimore, MD; #64927, Addgene; Zhou et al., 2015) and GFP-paxillin. HeLa cells were starved overnight in serum-free DMEM before images were acquired. Microscopy was performed using a Zeiss LSM 780 in online fingerprint mode to unmix the GFP and TORCAR biosensor signals. Activation of mTORC1, as visualized by the TORCAR biosensor, was monitored for an initial 3 min to measure baseline and then for a further 10 min after stimulation with FCS (final concentration, 10% FCS in DMEM). Data were analyzed offline using a bespoke MatLab script. Stable FAs were marked as regions of interest (ROIs), and those ROIs were transferred to analyze the biosensor signal. The same ROIs were shifted to a control region adjacent to GFP-paxillin–labeled FAs, and the biosensor signal was analyzed.

### Protein synthesis measurements
De novo protein synthesis was measured using Click-IT HPG Alexa Fluor 488 protein synthesis assay kit (Thermo Fisher Scientific) as per company instructions. Note, due to the kit requirements, the starvation–refeeding protocol differs from the standard protocol used elsewhere in the paper. Briefly, cells seeded on coverslips were subjected to serum starvation overnight followed by 1-h amino acid starvation. Cells were refed for 30 min with a mixture consisting of 1× leucine, 1× arginine, and 1× glutamine (to the concentrations found in standard DMEM) and supplemented with 100 nM insulin in the presence of the methionine analogue L-HPG, which is then incorporated into new proteins. Cells were then fixed and subjected to the HPG detection protocol as per company instructions.

### Generation of RPTOR constructs and cells
FLAG-RPTOR was fused at the C terminus to a 50-amino-acid peptide that corresponds to residues 979–1,028 in vinculin, the region required for the targeting of vinculin to FAs (KWSSKGNDIIAAAKRMALLMAEMSRLVRGGSGTKRALIQCAK

DIAKASDE; Wood et al., 1994) and named FLAG-RPTOR[FAT]. The sequence was produced by DNA synthesis and cloned into the pLV-puro plasmid by VectorBuilder, and stable HeLa cells expressing FLAG-RPTOR or FLAG-RPTOR[FAT] were produced via viral transduction and maintained in selection (1 µg/ml puromycin).

### Immunoprecipitations
mTOR and RPTOR immunoprecipitations were performed as previously described (Carroll et al., 2016). Briefly, cells were lysed in a buffer consisting of 40 mM Hepes, 2 mM EDTA, and 0.3% CHAPS supplemented with Halt protease and phosphatase inhibitors or 50 mM Hepes, 150 mM NaCl, 1 mM EDTA, and 1% NP-40 (for FLAG-RPTOR–paxillin immunoprecipitations; note, in this buffer, we were unable to detect RPTOR–mTOR binding). Cell lysates were incubated with prewashed protein A beads plus 3 µl/tube of anti-mTOR antibody or 20 µl (per 10-cm dish) magnetic FLAG antibody-coated beads for 2 h at 4°C with constant rotation. Beads were washed three times with lysis buffer. Proteins were eluted from the beads by incubation with 25 µl 0.2 M glycine-HCl, pH 2.5, for 10 min at room temperature. Eluent was neutralized by the addition of 2.5 µl Tris-HCl, pH 8.8. The samples were then mixed with sample buffer and boiled at 100°C for 5 min before being subjected to Western blot analysis.

### Quantification and statistical analysis
Cells were divided into two different regions to quantify peripheral and intracellular p-S6 staining. To separate these two regions in an unbiased manner, the cell perimeter was defined by thresholding equivalent saturated images and the cell area was scaled in 10% decrements using ImageJ (Starling et al., 2016). After background subtraction, cumulative integrated p-S6 density was represented. The most external region (corresponding to 10% of the cell area) generated using this approach was considered peripheral, whereas the sum of the internal regions (corresponding to 90% of the cell area) was defined as intracellular. Quantification using ImageJ was performed on 30–60 cells per condition in three independent experiments. Corrected total cellular fluorescence (CTCF) of the area of interest was calculated as CTCF = integrated density (IntDens) – (area of selected cell × mean fluorescence of background readings). Colocalization of markers in the external region of the cells (corresponding to 10% of the cell area as described above) was calculated using Imaris 7.0. The cellular distribution of paxillin and p-S6 was assessed using the "plot profile" function in ImageJ, using a line of 20 µm in length, measuring the signal intensity of each channel from a region just outside the cell and toward the nucleus. Line plots from different regions of the cell were overlaid around the point of maximum paxillin signal for each line, and at least 10 line profiles were used to generate each plot. For quantification of p-S6 and HPG signal in FAs and adjacent (control) areas, paxillin-labeled FAs were marked as ROIs and, after background subtraction, cumulative integrated p-S6 and HPG densities were analyzed. The same ROIs were transferred to a control region, adjacent to paxillin-labeled FAs, to analyze both p-S6 and HPG IntDens. Quantification of immunoblots was performed using ImageJ. Unless otherwise

stated, two-tailed, unpaired Student's *t* tests were performed on experimental data from at least three individual experiments.

## Online supplemental material

Fig. S1 shows spatial association between mTORC1 and FAs. Fig. S2 shows mTORC1 activation at the cell periphery. Fig. S3 shows activation of mTORC1 signaling in the vicinity of FAs. Fig. S4 shows growth factor activation at FAs. Fig. S5 shows that disruption of FAs or impairment of peripheral lysosomal distribution inhibit mTORC1 activation by nutrients. Video 1 shows TORCAR activity in FAs and adjacent regions upon FCS stimulation. Table S1 lists RPTOR-proximal proteins identified using BioID2 and label-free MS. Table S2 shows mTOR signaling– and mTORC1/2–associated proteins detected in the meta-adhesome. Table S3 shows lysosome-associated proteins and growth factor receptors detected in the meta-adhesome.

## Acknowledgments

We are grateful to E. Bennett for technical assistance, Alex von Kriegsheim (University of Edinburgh) for help with proteomics, Gerry Hammond (University of Pittsburgh) and Martin Humphries (University of Manchester) for helpful advice during the revisions of the manuscript, and the Newcastle Bioimaging Facility.

This work was funded by the Biotechnology and Biological Sciences Research Council (V.I. Korolchuk). O.D.K. Maddocks is funded by a Cancer Research UK Career Development Fellowship (C53309/A19702). A. Byron was funded by Cancer Research UK. J. Stingele is supported by the European Research Council (starting grant 801750 DNAProteinCrosslinks) and the Alfried Krupp von Bohlen und Halbach-Stiftung (Alfried Krupp Prize for Young University Teachers). E. Ponimaskin was funded through the Deutsche Forschungsgemeinschaft (fund PO 732) and the Lobachevsky University 5-100 academic excellence program. B. Carroll is supported by a British Skin Foundation Young Investigator Award (007/yi/17) and an Academy of Medical Sciences Springboard Award (SBF005\1130). This research was funded in whole or in part by the Wellcome Trust (218547/Z/19/Z awarded to B. Carroll). For the purpose of open access, the authors have applied a CC BY public copyright license to any author-accepted manuscript version arising from this submission.

B. Vanhaesebroeck is a consultant for Karus Therapeutics, iOnctura, and Venthera and has received speaker fees from Gilead. The remaining authors declare no competing financial interests.

Author contributions: Y. Rabanal-Ruiz performed cell biology experiments and bioimaging analyses; A. Byron and J.C. Wills performed proteomic and bioinformatic analyses; A. Wirth and E. Ponimaskin carried out TORCAR experiments and analysis; R. Madsen, A. Zeug, and B. Vanhaesebroeck assisted with analyses of growth factor activation; L. Sedlackova performed cell biology experiments; G. Hewitt and J. Stingele generated constructs and cell lines; G. Nelson supervised and performed bioimaging experiments; T. Zhang and O.D.K. Maddocks performed metabolomic analyses; R. Fässler provided materials; B. Carroll performed BioID and cell biology experiments; Y. Rabanal-Ruiz, A. Byron, A. Wirth, O.D.K. Maddocks, E. Ponimaskin, B. Carroll, and V.I. Korolchuk designed and supervised elements of the study; B. Carroll, and V.I. Korolchuk supervised the entire project; Y. Rabanal-Ruiz, A. Byron, B. Carroll, and V.I. Korolchuk wrote the manuscript with help from all authors.

Submitted: 2 April 2020

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

# Supplemental material

Rabanal-Ruiz et al.
mTORC1 and focal adhesions

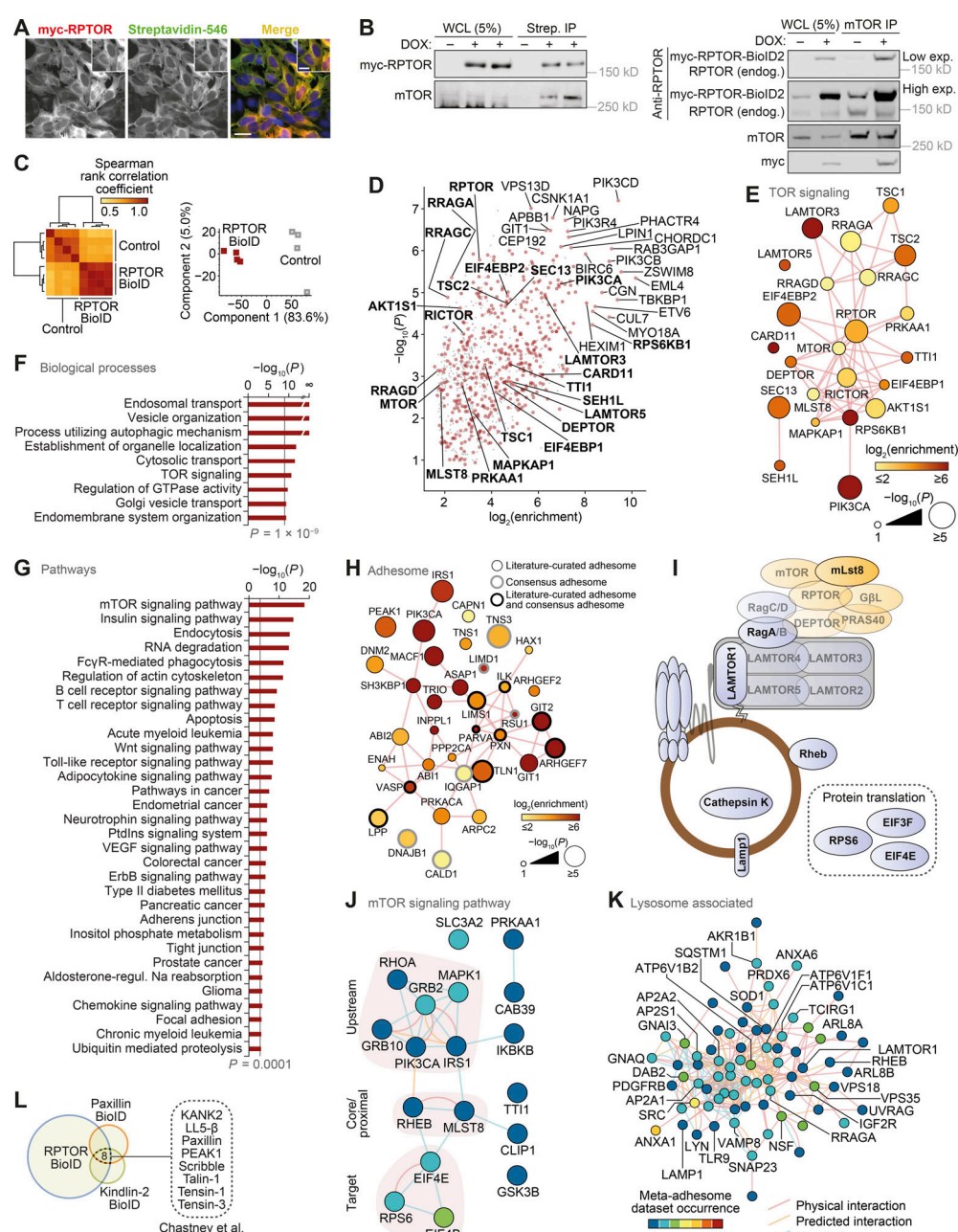

**Figure S1. RPTOR BioID2 reveals spatial association between mTORC1 and FAs. (A)** Immunostaining of RPTOR-BioID2-expressing U2OS cells treated with 50 µM biotin overnight. Cells were stained for biotin and myc-tagged RPTR. Scale bars, 20 µm (insets, 10 µm). Nuclei were visualized with DAPI. **(B)** Representative immunoblots of streptavidin pull-downs following 50 µM biotin incubation overnight (left) and immunoprecipitation (IP) with anti-mTOR antibody (right). WCL, whole-cell lysate. **(C)** Unsupervised analysis of RPTOR BioID samples (BioID2-based proximity labeling) quantified by MS. Hierarchical cluster analysis of Spearman rank correlation coefficients of pairwise comparisons for all samples analyzed by MS (P < 2 × 10$^{-127}$, Spearman's test; left). Principal-component analysis of samples analyzed by MS (right). The first two principal components account for 88.6% of the total variance of the dataset. **(D)** Enrichment of noncontaminant proteins specifically enriched in RPTOR BioID samples ($n$ = 4 independent experiments; P < 0.01, two-sided Welch's $t$ test with permutation-based FDR correction, artificial within groups variance = 2). Proteins enriched by at least 256-fold or with P < 10$^{-6}$ (two-sided Welch's $t$ test) are labeled; enriched TOR signaling components (Gene Ontology accession 0031929) are labeled in bold. Putative contaminant proteins are indicated by gray crosses. **(E)** Interaction network analysis of TOR signaling components identified in U2OS cells by RPTOR BioID and MS. **(F and G)** KEGG pathway gene set enrichment analysis and Gene Ontology biological process overrepresentation enrichment analysis of noncontaminant proteins specifically enriched in RPTOR BioID samples. Biological processes enriched with P < 10$^{-9}$ (hypergeometric test with Benjamini–Hochberg correction; F) and pathways enriched with P < 0.0001 (Kolmogorov–Smirnov test with FDR correction; G) are shown. **(H)** Interaction network analysis of RPTOR-proximal adhesome components identified by RPTOR BioID. **(I)** Diagrammatic representation of mTORC1 and lysosomal proteins identified in meta-adhesome datasets. See Table S2 and Table S3 for more details. **(J and K)** Interaction network analysis of reported physical and predicted protein–protein interactions and pathway associations between mTOR signaling pathway components (J) and lysosome-associated proteins (K) detected in adhesion complex proteomes (meta-adhesome datasets). See Table S2 and Table S3 for additional details. **(L)** Identification of adhesion proteins significantly enriched in the RPTOR BioID2 dataset that were also identified in published BioID datasets of the FA proteins paxillin and kindlin-2 in mouse pancreatic fibroblasts (Chastney et al., 2020).

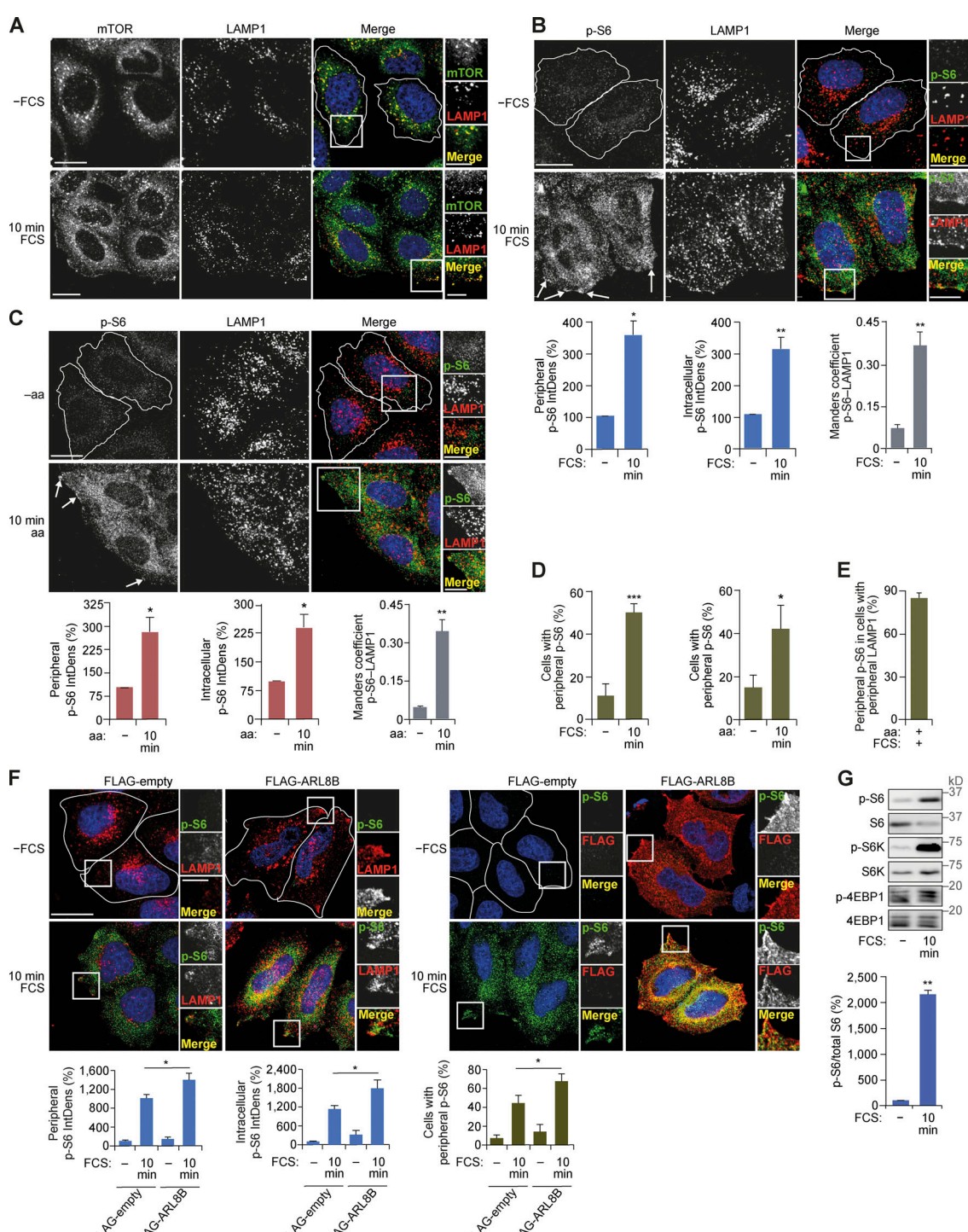

Figure S2. **mTORC1 is activated at the cell periphery. (A)** HeLa cells grown in full-nutrient medium were FCS starved for 18 h (−FCS) or FCS starved and then recovered in FCS-containing medium for 10 min (10 min FCS). Immunostaining of mTOR and LAMP1 is shown. **(B and C)** Immunostaining for p-S6 and LAMP1 following starvation and refeeding as in A (B) or in response to amino acid starvation versus amino acid starvation and recovery (C). Arrows indicate mTORC1 activation at the cell periphery. The IntDens of peripheral or intracellular signal of p-S6 were quantified and the colocalization at the cell periphery between p-S6 and LAMP1 was quantified using Manders coefficient. **(D)** The proportion of cells with peripheral p-S6 staining was quantified in cells starved and recovered in FCS-containing medium (left) or amino acids (right) as indicated. **(E)** Quantification of the proportion of cells with peripheral LAMP1 in full-nutrient medium that exhibited peripheral p-S6 staining. **(F)** HeLa cells transfected with FLAG-ARL8B or empty FLAG plasmid were FCS starved for 18 h (−FCS) or FCS starved and then recovered in FCS-containing medium for 10 min (10 min FCS). Immunostaining of p-S6 and LAMP1 (left) or FLAG (right) is shown. The IntDens of peripheral or intracellular signal of p-S6 and the proportion of cells with peripheral p-S6 staining was quantified. **(G)** Cells were treated as in A, lysed, and subject to immunoblotting for mTORC1 activity (top). For quantification of relative p-S6 levels (bottom), error bars represent SEM; n = 3 independent experiments (for IntDens, n ≥ 10 cells were quantified per experiment). *, P < 0.05; **, P < 0.01; ***, P < 0.001; two-sided Student's t test. Scale bars, 20 μm (insets, 10 μm). Nuclei were visualized with DAPI.

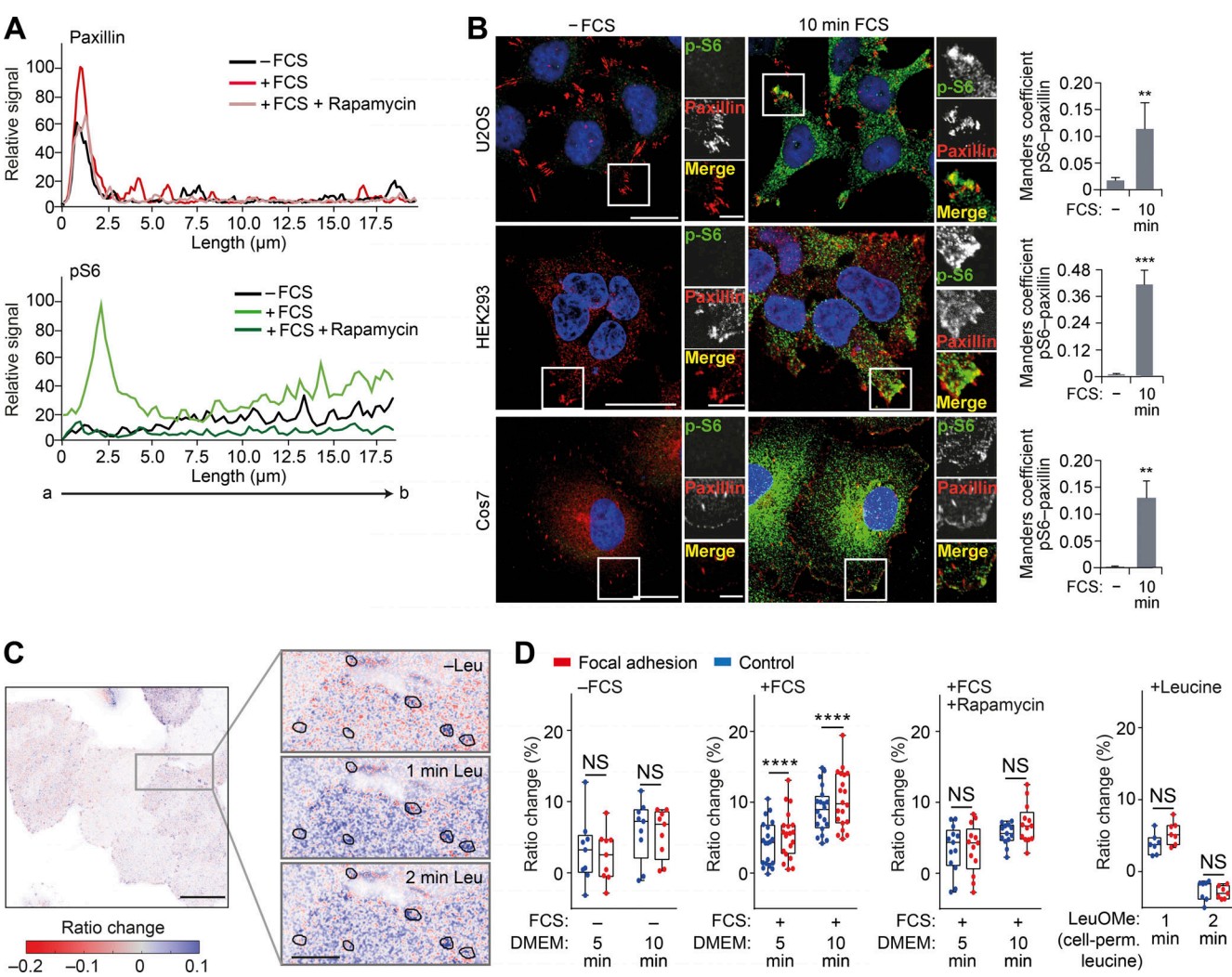

Figure S3. **mTORC1 signaling is activated in the vicinity of FAs. (A)** Fluorescence intensity line profile plots corresponding to lines exemplified by an arrow in Fig. 3 A. **(B)** U2OS, HEK293, and Cos7 cells grown in full-nutrient medium were FCS starved for 18 h (−FCS) or FCS starved and then recovered in FCS-containing medium for 10 min (10 min FCS) and immunostained for p-S6 and the FA protein paxillin. Colocalization (Manders coefficient) between p-S6 and paxillin was analyzed. **(C)** Representative confocal image of GFP-paxillin intensity-scaled ratio image of TORCAR FRET-based biosensor in serum-starved HeLa cells before and after stimulation with 25 mM membrane-permeable leucine methylester. Note that, to visualize the differences, the images were gamma adjusted, and black borders in zoom insets highlight the ROIs corresponding to FAs. **(D)** Quantification of ratio change (percentage) of TORCAR biosensor signal before stimulation (−FCS), after stimulation with FCS (+FCS), in the presence of rapamycin (+FCS +Rapamycin), and in response to leucine methylester (+Leucine). Each data point represents a coverslip including several cells; box-and-whisker plot whiskers represent minimum and maximum values. Error bars represent SEM; $n$ = 3 independent experiments. ****, $P < 0.0001$; two-way ANOVA with Sidak correction. For B, **, $P < 0.01$, ***, $P < 0.001$ two-sided Student's $t$ test. Scale bars, 20 µm (insets, 10 µm). Nuclei were visualized with DAPI.

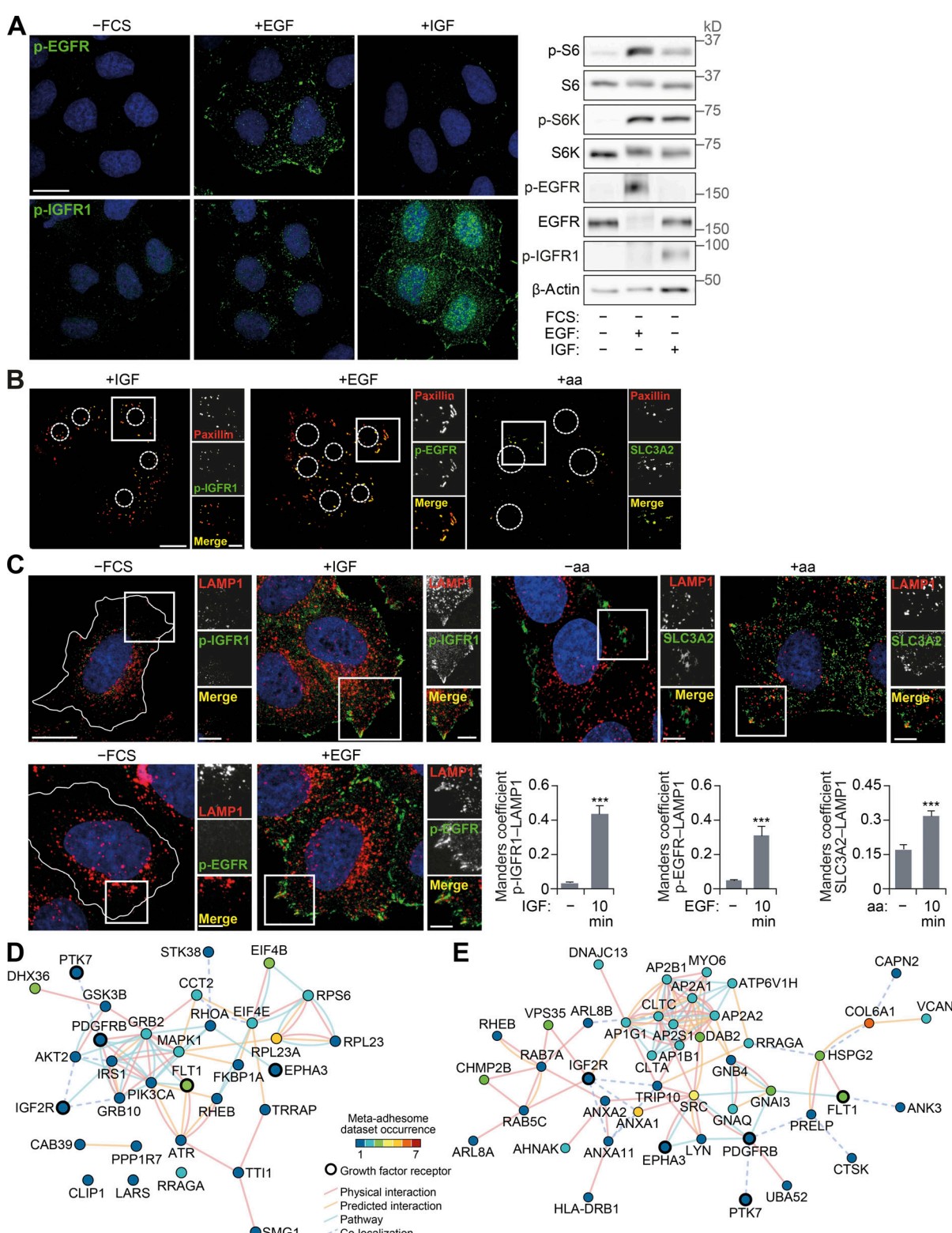

Figure S4. **Growth factor activation at FAs. (A)** HeLa cells FCS starved for 18 h (−FCS) or starved and stimulated with EGF or IGF for 10 min were analyzed by immunofluorescence (left) and immunoblotting (right). **(B)** Cells were starved and stimulated with IGF, EGF, or amino acids. FAs remaining on coverslips after removal of cell bodies were immunostained using paxillin and p-IGFR1, p-EGFR, or SLC3A2 antibodies. **(C)** Cells were starved and stimulated as indicated and immunostained for p-EGFR, p-IGFR1, SLC3A2, and LAMP1, and colocalization at the cell periphery (Manders coefficient) was quantified. **(D and E)** Interaction network analysis of growth factor receptors and mTOR-associated proteins (D) and lysosome-associated proteins (E) detected in adhesion complex proteomes (meta-adhesome datasets). Proteins (nodes) are annotated with gene names for clarity. Error bars represent SEM; n = 3 independent experiments. Scale bars, 20 µm (insets, 10 µm). Nuclei were visualized with DAPI; original localization of nuclei in FA preparations (B) is indicated by dashed circles. For C, ***, P < 0.001; two-sided Student's t test.

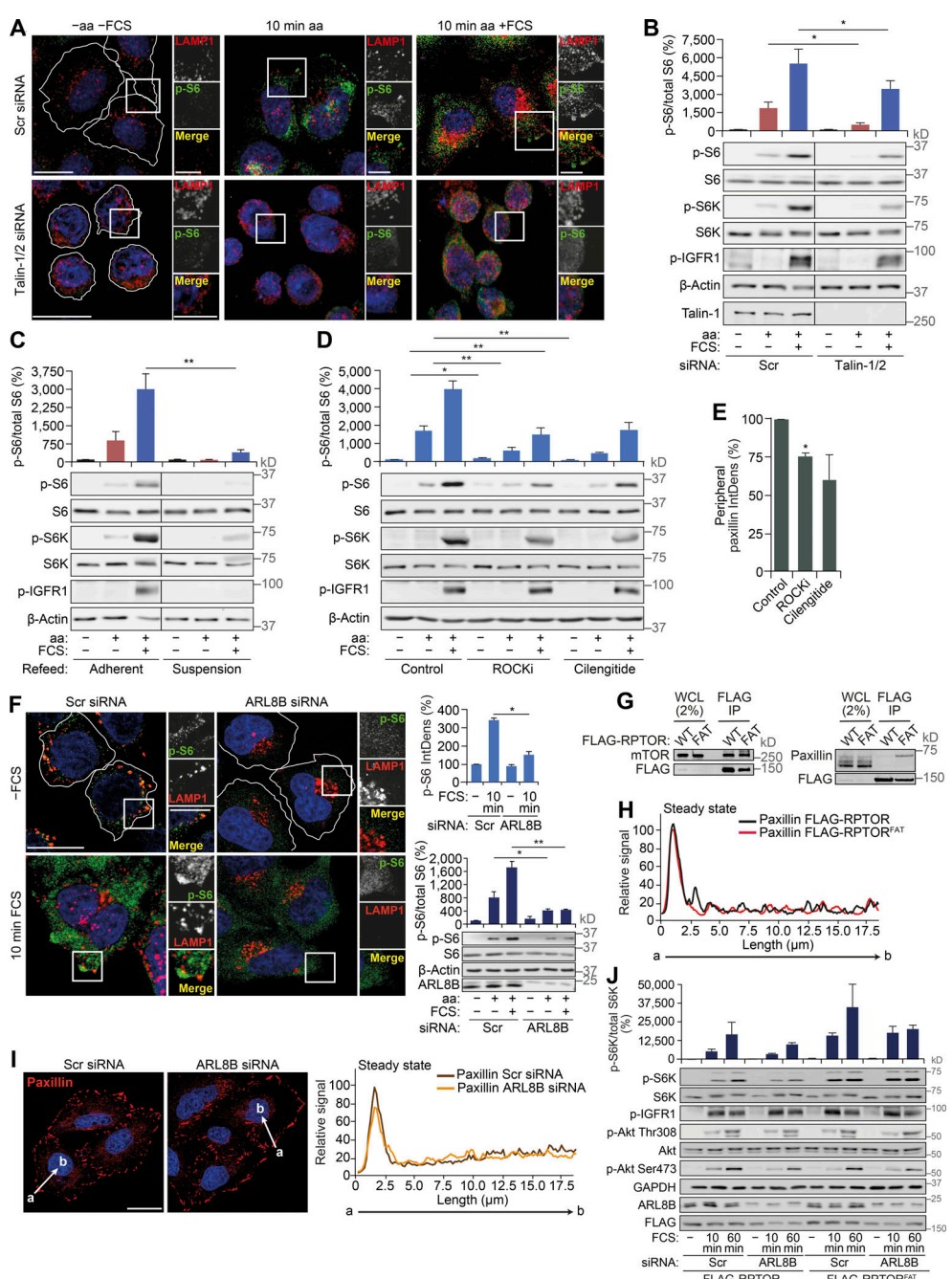

Figure S5. **Disruption of FAs or impairment of peripheral lysosomal distribution inhibits mTORC1 activation by nutrients. (A and B)** HeLa cells transfected with scrambled (Scr) or TLN1/2 siRNAs grown in full-nutrient medium were amino acid and FCS starved for 1 h (−aa −FCS) and then recovered in amino acid–containing medium (10 min aa) or amino acid– and FCS-containing medium (10 min aa +FCS). Cells were analyzed by immunostaining for LAMP1 and p-S6 (A) or immunoblotting to detect mTORC1 activity (B). **(C)** HeLa cells subjected to amino acid and FCS starvation for 1 h were scraped (suspension) or left adherent and then refed as described in A and subjected to immunoblot analysis to detect mTORC1 activity. **(D and E)** HeLa cells were FCS starved for 18 h (−FCS); treated with DMSO (control), ROCKi, or integrin antagonist (cilengitide) for 1 h in −FCS medium; starved of amino acids (−aa −FCS) for 1 h in the presence of inhibitors; and then recovered in full-nutrient medium for 10 min. Cells were subjected to immunoblot analysis to detect mTORC1 activity (D). The IntDens of peripheral paxillin staining was quantified (E). **(F)** Scrambled (Scr) or ARL8B siRNA–transfected HeLa cells were FCS starved for 18 h (−FCS) and then recovered in FCS-containing medium for 10 min (10 min FCS). Immunostaining of p-S6 and LAMP1 is shown. **(G)** HeLa cells expressing FLAG-RPTOR and FLAG-RPTOR$^{FAT}$ grown in full-nutrient media were lysed, immunoprecipitated with FLAG antibody, and immunoblotted for FLAG and mTOR (left) and FLAG and paxillin (right). **(H)** Fluorescence intensity line profile plots corresponding to lines exemplified by arrows in Fig. 6 A are shown. **(I)** Scrambled or ARL8B siRNA–transfected HeLa cells were FCS starved for 18 h (−FCS) and then recovered in FCS-containing medium for 10 min (10 min FCS). Immunostaining for paxillin (left) and fluorescence intensity line profile plots corresponding to lines exemplified by arrows (right) are shown. **(J)** Scrambled or ARL8B siRNA–transfected HeLa cells expressing FLAG-RPTOR or FLAG-RPTOR$^{FAT}$ grown in full-nutrient medium were FCS starved for 18 h (−FCS) or FCS starved and then recovered in FCS-containing medium for 10 or 60 min (10, 60 min FCS), lysed, and subjected to immunoblot analysis using antibodies as shown. Error bars represent SEM; n = 3 independent experiments. For J, n = 2 independent experiments and error bars represent SD. *, P < 0.05; **, P < 0.01; two-sided Student's t test (performed between groups). Scale bars, 20 µm (insets, 10 µm). Nuclei were visualized with DAPI.

Video 1.   **TORCAR activity in FAs and adjacent regions upon FCS stimulation.** Representative confocal time-lapse video of GFP-paxillin intensity-scaled ratio image of TORCAR FRET-based biosensor in serum-starved HeLa cells (left). The movie highlights the normalized ratio change as well as the intensity of the biosensor in different parts of the cells during a 13-min measurement (80 frames in total, 10-s interval). The traces of the normalized ratio change (right) are shown with the same color coding as the ROIs in the image. Cold-colored ROIs correspond to FAs and warm-colored ROIs to control areas (adjacent to FAs). The cells were treated after 3 min with 10% FCS until the end of the measurement. Upon stimulation, the normalized ratio within the ROIs showing FAs decreases strongly, which is visualized by red color. The quantification of such measurements is shown in Fig. 3 and Fig. S3.

**Three tables are provided online as separate Excel files. Table S1 lists RPTOR-proximal proteins identified using BioID2 and label-free MS. Table S2 lists mTOR signaling– and mTORC1/2-associated proteins detected in the meta-adhesome. Table S3 lists lysosome-associated proteins and growth factor receptors detected in the meta-adhesome.**

