## [Peer Review File · The Journal of Cell Biology]

mTORC1 activity is supported by spatial association with focal adhesions

Yoana Rabanal-Ruiz, Adam Byron, Alexander Wirth, Ralitsa Madsen, Lucia Sedlecova, Graeme Hewitt, Glyn Nelson, Julian Stingele, Jimi Wills, Tong Zhang, Andre Zeug, Reinhard Faessler, Bart Vanhaesebroeck, Oliver Maddocks, Evgeni Ponimaskin, Bernadette Carroll, and Viktor Korolchuk

Corresponding Author(s): Bernadette Carroll, School of Biochemistry, University Walk, Bristol BS8 1TD and Viktor Korolchuk, Newcastle University

Review Timeline:	Submission Date:	2020-04-02
	Editorial Decision:	2020-05-01
	Revision Received:	2020-11-21
	Editorial Decision:	2020-12-31
	Revision Received:	2021-01-29

Monitoring Editor: Harald Stenmark

Scientific Editor: Melina Casadio

Transaction Report:

DOI: <https://doi.org/10.1083/jcb.202004010>

May 1, 2020

Re: JCB manuscript #202004010

Dr. Bernadette Carroll
University of Bristol
Biomedical Sciences Building
Bristol BS8 1TD
United Kingdom

Dear Dr. Carroll,

Thank you for submitting your manuscript entitled "mTORC1 activity is supported by spatial association with focal adhesions". Your manuscript has been assessed by expert reviewers, whose comments are appended below. Although the reviewers express potential interest in this work, significant concerns unfortunately preclude publication of the current version of the manuscript in JCB.

You will see that it's clear that your model connecting mTORC1 activity to focal adhesions (FAs) spatially is interesting to adhesion and mTOR experts. Based on their interest, which we share, we'd like to consider the work further. However, the referees raise similar, significant issues with the work: the evidence suggesting that focal adhesions promote mTORC1 signaling and are sites of mTORC1 activation is not fully convincing because you don't rule out other factors that could contribute to the phenotypes seen (e.g., upon Arl8B depletion, Rev#1 #5 and Rev#3; upon impairment of FA proteins, per Rev#3) and the work lacks the necessary rescues and controls to show specific effects of adhesions on mTORC1 signaling (see Rev#1 main comment and points #2 - lack of specificity in Ab used; #3/4 lack of rapamycin control; #6; Rev#2 #3-4-5). The referees additionally comment on the multiple systems used (Rev#1, Rev#2 #2). In addition, their remarks suggest that the results are not sufficiently well integrated with the literature (e.g., Rev#2 points #6 and #1) - including prior work linking mTOR to fibrillary adhesions as stressed by Rev#2. Both Revs #1 and #3 have questions about the hits from the proteomics studies (Rev#1 #1 and Rev#3 minor point #1).

We have editorially discussed these issues and find them valid and important. Because there are a number of factors that could contribute to the effects of FAs on mTORC, as the reviewers highlighted with great detail, definitive evidence is needed to implicate FAs or particular types of adhesions in promoting peripheral mTORC1 activation. More work is needed to provide much stronger and definitive evidence that should convince experts in the fields of metabolism and adhesion for publication.

Please let us know if you are able to address the major issues outlined above and wish to submit a revised manuscript to JCB. Note that a substantial amount of additional experimental data likely would be needed to satisfactorily address the concerns of the reviewers in full. It may be necessary to extend your manuscript to a full Research Article. As you may know, the typical timeframe for revisions is three to four months. However, we at JCB realize that the implementation of social distancing and shelter in place measures that limit spread of COVID-19 also pose challenges to scientific researchers. Lab closures especially are preventing scientists from conducting

experiments to further their research. Therefore, JCB has waived the revision time limit. We recommend that you reach out to the editors once your lab has reopened to decide on an appropriate time frame for resubmission. Please note that papers are generally considered through only one revision cycle, so any revised manuscript will likely be either accepted or rejected.

If you choose to revise and resubmit your manuscript, please also attend to the following editorial points. Please direct any editorial questions to the journal office.

GENERAL GUIDELINES:

Text limits: Character count for a Report is < 20,000; a full Research Article is < 40,000, not including spaces. Count includes title page, abstract, introduction, results, discussion, acknowledgments, and figure legends. Count does not include materials and methods, references, tables, or supplemental legends.

Figures: A Report may include up to 5 main text figures; a full Research Article may have up to 10 main text figures. To avoid delays in production, figures must be prepared according to the policies outlined in our Instructions to Authors, under Data Presentation, <http://jcb.rupress.org/site/misc/ifora.xhtml>. All figures in accepted manuscripts will be screened prior to publication.

IMPORTANT: It is JCB policy that if requested, original data images must be made available. Failure to provide original images upon request will result in unavoidable delays in publication. Please ensure that you have access to all original microscopy and blot data images before submitting your revision.

Supplemental information: There are strict limits on the allowable amount of supplemental data. Reports may have up to 3 supplemental figures; a full Research Article may have up to 5 supplemental figures. Up to 10 supplemental videos or flash animations are allowed. A summary of all supplemental material should appear at the end of the Materials and methods section.

If you choose to resubmit, please include a cover letter addressing the reviewers' comments point by point. Please also highlight all changes in the text of the manuscript.

Regardless of how you choose to proceed, we hope that the comments below will prove constructive as your work progresses. We would be happy to discuss them further once you've had a chance to consider the points raised. You can contact the journal office with any questions, cellbio@rockefeller.edu or call (212) 327-8588.

Thank you for thinking of JCB as an appropriate place to publish your work.

Sincerely,

Harald Stenmark, PhD
Monitoring Editor, Journal of Cell Biology

Melina Casadio, PhD
Senior Scientific Editor, Journal of Cell Biology

Reviewer #1 (Comments to the Authors (Required)):

The manuscript by Rabanal-Ruiz et al proposes an intriguing hypothesis that localization of mTORC1 near the focal adhesions (FA) at the plasma membrane brings mTORC1 activity "into close association with nutrient inputs". Specifically, the authors claim that (1) "mTORC1 is activated in specific regions colocalizing with FAs" and that (2) "FAs are required for maximal mTORC1 activation in response to nutrients". To support their claims, the authors present the following evidence:

- Nearly half of proteins identified in a BioID experiment with an mTORC1 component Raptor overlap with proteins identified in FAs
- mTOR, LAMP1 and paxillin show partial FCS/aa-induced colocalization
- in about half of cells, IF of phosphorylated mTORC1 substrates 4EBP1 and S6 shows increase both in the cell periphery and interior, partially overlapping with LAMP1 and paxillin staining
- a ratiometric FRET reporter for the mTORC1-specific substrate 4EBP1 indicates lower FRET in sites overlapping with GFP-paxillin when treated with FCS
- KD of ARL8B leading to perinuclear lysosome accumulation decreases phosphorylation of mTORC1 substrate S6
- Talin1/2 KO cells show reduced S6 phosphorylation (both in the periphery and in cell interior)
- ROCK-specific inhibitor and integrin antagonist reduced S6 phosphorylation
- Cells with constitutive mTORC1 signaling show increased number of FAs

Overall, the presented evidence points in favor of the authors' hypothesis, albeit indirectly. I personally find the hypothesis is indeed interesting, generally in line with previous reports and worth publishing; yet, the presented evidence is mostly correlative and insufficient to examine the cause-and-effect relationship. While some of the authors' models and tools (e.g., Talin1/2 KO cells, Arl8B KD, ratiometric FRET reporter) appear to be well-suited for examining their working hypothesis in detail, the key experiments, such as Talin- or Arl8B rescue, inhibition of mTORC1/mTOR with rapamycin/Torin1, Akt inhibition, using FRET reporter in Talin1/2 KO cells, etc, are missing. I'd be supporting publication of the report, but in my opinion it would require major revision. I'd recommend the authors to clarify the working hypothesis (what does "SUPPORTING mTORC1 activity at the periphery" exactly mean?) and concentrate on 1-2 model systems.

Major comments:

1. As mTORC1 could be both membrane-bound and cytosolic, I'd expect a lot of proteins to be biotinylated by a cytosolic protein over 16 hrs. The labeling radius of 10 nm the authors refer to was estimated for a stable transmembrane complex (NPC) and therefore may not necessarily apply here. Further, from the presented data it is not clear what fraction of BioID2-fused Raptor is integrated into a complex. As the enzyme could potentially interfere with mTORC1 formation, this estimation is very relevant for the BioID/MS results. I'd expect some sort of negative control to be quite useful to demonstrate labeling specificity. A statement on the reproducibility of the proteomic data would also be desirable: the authors indicate that bead-bound proteins from four independent experiments we used; were they pooled or analyzed separately?
2. The authors use mTOR-specific antibody as a proxy for mTORC1. Given that mTOR is a part of both mTORC1 and mTORC2, and at least a significant portion of mTORC2 localizes to the PM, Raptor-specific Ab appears to be a better choice for IF.
3. The authors indicate that >40% of cells show phospho-S6 staining at the periphery. What happens to the other cells? Do they not have peripheral phospho-S6 or do not respond to re-feeding? Further, in such experiments, rapamycin treatment is a natural control needed to establish

mTORC1 specificity; why didn't the authors use it?

4. I am somewhat worried about the results obtained using TORCAR FRET reporter, which appears to be one of the strongest evidence for the authors' conclusions. First, rapamycin should be used after FCS treatment to demonstrate both the reversibility and mTORC1 specificity of the FRET signal. Loss of FRET is often associated with unequal photobleaching, especially with a relatively low dynamic range of the FRET signal. Next, GFP and YPet excitation spectra overlap significantly, which could result in weird ratio even after unmixing; the lack of the signal in the "control" (i.e., GFP-free) areas could equally well be interpreted as an unmixing artifact. I'd strongly recommend using far-red FPs for paxillin. Finally, I could not find an example of the FRET ratio image when cells are treated with cell-permeable leucin analog; please add.

5. Arl8b KO was shown to significantly delay mTORC1 as well as mTORC2/Akt activity upon re-feeding, with ~30-40% recovered only after 1 hour (Jia and Bonifacino, Mol Cell 2019). The authors, however, compare phospho-S6 staining only after 10 min. Does the peripheral mTORC1 pool still contribute to S6 phosphorylation 1 hr after re-feeding? Could PM-localized Akt (and not FAs) play a role in mTORC1 activation at the PM (to remove Tsc2)?

6. Experiments in Talin1/2 KO cells, adherent/suspension cells and with ROCKi and integrin antagonist are the strongest evidence supporting the authors' hypothesis, but they require proper specificity controls with mTORC1, mTORC2, PI3K and Akt substrates and the corresponding rescue/inhibitor washout experiments wherever possible.

7. Finally, I think it is important to better define the authors' working hypothesis. They propose that "peripheral dispersal of mTORC1-positive lysosomes supports mTORC1 activation by bringing it into close proximity to incoming mitogenic signaling cascades and amino acids". Of course this could be the case, but discrimination between the contribution of the peripheral and interior pools of mTORC1 seems to be a non-trivial task, as such spatially distinct pools could have different catalytic activities, flux of substrates or distinct inactivation kinetics. Unraveling the contribution of these parameters would require careful experimentation.

The reviewer cannot comment on the quality of the proteomic data (out of expertise). The reviewer claims no competing interests.

Reviewer #2 (Comments to the Authors (Required)):

In the present study the authors expand on their earlier exciting data showing that peripheral lysosomes are a feature of fed cells and that starvation triggers perinuclear lysosome localisation. Here the authors have exploited the powerful proximity-biotinylation method and find that adhesion components are significantly overrepresented in the RPTOR BioID dataset. They go on to show that mTOR and lysosomes are tethered to focal adhesion, not directly through FA components but via FA associated growth factor receptors. This is a predominantly carefully conducted study. However, I have some suggestions that the authors may want to consider addressing to further increase the impact of the findings and the relevance to the adhesion biology field.

1) The authors provide a nicely illustrated and informative representation of their RPTOR BioID2 dataset that convincingly demonstrates the proximity of mTORC1 with focal adhesions. Interestingly, two of the nine proteins were found significantly enriched in RPTOR/Kindlin/paxillin proximity proteomes are linked predominantly with fibrillar adhesions (tensin-1 and tensin-3) and two are components of the FA proximal structures that link focal adhesion to microtubules (Kank and liprin). These data would imply that mTORC1 proximity is not linked to focal adhesion specifically but possibly engages different types of adhesions in cells. This would also be in line with

a previous study showing recruitment of lysosomes and active mTOR to fibrillary adhesions (DOI:<https://doi.org/10.1016/j.celrep.2014.12.037>). Could the authors expand on their data in Figure 2 to also other adhesion types than paxillin-positive adhesions?

2) The proximity biotinylation was carried out in osteosarcoma U2OS cells. Why is the validation data of mTOR and LAMP1 recruitment to paxillin positive adhesion with HeLa cells? The authors should consider showing this (and possibly localization to other adhesion types as well) in other cell types to demonstrate the generality/specificity of these data (there is some data from MEFs and MRC5 cells later, but would be nice to have a broader validation here).

3) Figure 3a. Could the authors somehow demonstrate that the "significant increase in co-localisation between p-S6 and paxillin" is not just a reflection of having low p-S6 in the cell in starved conditions to having it abundantly throughout the cells in fed conditions?

4) Figure 4a. How have the authors validated the specificity/ruled out off-target effects of the ARL8B siRNA? Does ARL8B siRNA influence adhesion number or distribution in the cells. Starvation in other cell models has been shown to impact on the balance between peripheral focal and adhesions and elongated adhesions localising under the cell body.

5) Figure 4G would be easier to interpret if starved and fed condition were shown like in the other experiments.

6) In the discussion the authors speculate that "Activation of mTORC1 at FAs by nutrients may therefore suppress autophagy of FA components". They should mention earlier work describing already that mTOR activity suppressed integrin and ECM protein uptake into spatially localised adhesion proximal lysosomes.

Minor points:

Figure 2c. What is SDHA? Why was it chosen as control? It might be better to score the number of PLA spots/cell or the number of PLA spots overlapping with Paxillin rather than the % of PLA positive cells.

Figure 3c would benefit from adding labelling in the images (to indicate what the green signal is). Page 7 top, this sentence may need to be edited "although at this point is cannot necessarily be completely ruled out."

Reviewer #3 (Comments to the Authors (Required)):

The present work provides evidence that focal adhesions are required for maximal mTORC1 activation in response to nutrients. The authors based investigations on their earlier work demonstrating that the localization of lysosomes near the plasma membrane increases mTORC1 activity by bringing it into close association with nutrient inputs into the cell.

Here it is shown that mTORC1 interacts with focal adhesion proteins by several proteomics mass spectrometry-based approaches that were confirmed and analyzed in detail by complementary immunofluorescence approaches. Although data analyses were performed at a state-of-the-art level, and the results are clearly represented the manuscript falls short in demonstrating under which physiological conditions the postulated "maximal mTORC1 activation" would be required. In addition, although in the Arl8b KD experiment mTORC1 signaling is compromised, it remains to

my opinion unclear if the results obtained can be really assigned to a lack of targeting focal adhesions. It could be equally well possible that Arl8b KD might interfere with mTORC1 signaling by increasing the amount of regulator associating with BORG and thereby sequestering it away from Rags. If one took this into account, the conclusions would be completely different. Therefore, the authors could for instance artificially target mTORC1 to FA (eg. FAT domain on Rheb) and look if this would rescue the signaling defect detected upon Arl8b KD.

Importantly, depletion of focal adhesions, either by Talin KD or ROCK inhibitors, has a significant impact on the cytoskeleton. Chemotactic changes and/or alterations in Rac1 activity have previously been shown to regulate both mTORC1 and mTORC2. Therefore, presented experiments should be interpreted with caution since some of the effects observed might be independent of the here described focal adhesion dependent activation of mTORC1.

Minor points that should be revisited by the authors:

Figure 1: The results from the MS analysis of the BioID2-RPTOR pulldowns is in part surprising. Why is the log₂ enrichment for the autophagy related proteins considerably higher than that observed for core mTORC1 subunits?

Figure 3: The authors state that "FAs play an important role in the early events of mTORC1 activation in response to nutrients". Unfortunately, there are a few points unclear to this reviewer concerning the Leucine methyl ester stimulation experiment. In the legend from figure 3 it is mentioned that in panel E, "each data point represents a coverslip including several cells" whereas in panel F, "each data point represents a single cell". Also, in the legend from supplementary figure 3 panel A, "each data point represents a coverslip including several cells". Could the authors please explain why the analysis from the Leucine methyl ester stimulation was apparently performed differently from the remaining TORCAR biosensor experiments? Further, it would make the authors claim stronger if they could include an independent positive control, e.g. immunoblotting, to detect mTORC1 activation upon Leucine methyl ester stimulation.

Supplementary Figure 2: The image quality on subcellular localization of p-4EBP1 after AA refeeding is not convincing. The entire Panel would also profit from colocalization analysis with either LAMP1 or Paxillin, similarly to what is provided in the remaining panels of the same figure.

Figure 4b shows quantified integrated densities of pS6 signals in IF experiments. Somewhat surprisingly the decrease of pS6 signal upon ARL8B knock-down is almost the same for peripheral and intracellular pS6. Such data presentation might give an impression to the readers that peripheral (10% of the cell area) vs intracellular regions were not properly defined. Considering an approximately 2.5-fold decrease in total cell number with peripheral p-S6 upon ARL8b depletion (panel c), one would expect a more specific effect for peripheral staining. The same criticism applies to corresponding Supplementary Figures 4b-c.

Corresponding to panels a-c in Figure 4, panel d shows immunoblotting analyses of total pS6. It is partially misleading because the conditions (aa and aa plus FCS) are different from panels a-c (FCS starvation and FCS stimulation), and more related to Supplementary Figures 4a-d. The exact conditions are not described, but one could assume 18 h FCS starvation (panels a-c) and 1h aa/FCS starvation in panel d.

Dear Editors,

We appreciate the constructive criticism and suggestions made by all three reviewers on our manuscript entitled "mTORC1 activity is supported by spatial association with focal adhesions". We have now performed extensive revisions and are pleased to offer the modified manuscript for your consideration. Please also find below a point-by-point response to the reviewers' comments, where our replies are in red.

Reviewer #1 (Comments to the Authors (Required)):

Major comments:

1. As mTORC1 could be both membrane-bound and cytosolic, I'd expect a lot of proteins to be biotinylated by a cytosolic protein over 16 hrs. The labeling radius of 10 nm the authors refer to was estimated for a stable transmembrane complex (NPC) and therefore may not necessarily apply here. Further, from the presented data it is not clear what fraction of BioID2-fused Raptor is integrated into a complex. As the enzyme could potentially interfere with mTORC1 formation, this estimation is very relevant for the BioID/MS results.

We agree that soluble cytosolic proteins could potentially interact with significantly more molecules than transmembrane components. However, we could not find any evidence from the literature that this would be the case, and the labelling radius for BioID2 would nevertheless still be expected to remain the same. Therefore, we added an acknowledgement in the text that the proximity is only an estimate from the studies with NPC components. At the same time, as suggested, we performed additional controls for our BioID2-fused RPTOR construct. Specifically, we show that RPTOR-BioID2 biotinylates mTOR and is efficiently incorporated into mTORC1 complex (Fig. S1 B).

I'd expect some sort of negative control to be quite useful to demonstrate labeling specificity. A statement on the reproducibility of the proteomic data would also be desirable: the authors indicate that bead-bound proteins from four independent experiments we used; were they pooled or analyzed separately?

A negative control was included in all four independent experimental replicate analyses (see Fig. S1 C and Table S1). The four independent experiments were not pooled, so we were able to assess reproducibility (Fig. S1 C and Table S1 of the manuscript). All four independent experiments were highly correlated (RPTOR BioID Spearman rank correlation coefficients > 0.91; negative control Spearman rank correlation coefficients > 0.81; Fig. S1 C). We have now included a statement of reproducibility and clarified the specific experimental conditions that were analysed.

2. The authors use mTOR-specific antibody as a proxy for mTORC1. Given that mTOR is a part of both mTORC1 and mTORC2, and at least a significant portion of mTORC2 localizes to the PM, Raptor-specific Ab appears to be a better choice for IF.

The main finding of the paper is that mTORC1 is activated in the vicinity of FAs, thus we use phospho-S6, which is specific to mTORC1, not mTORC2. Additionally, we use the mTORC1-specific FRET reporter TORCAR. As there are no reagents that would allow us to characterise intracellular localisation of RPTOR, we cannot carry out the proposed experiment; however, we made it clear in the text that both phospho-S6 and TORCAR are the measure of mTORC1 but not mTORC2 activity and included rapamycin controls for all our assays to demonstrate their specificity for mTORC1.

3. The authors indicate that >40% of cells show phospho-S6 staining at the periphery. What

happens to the other cells? Do they not have peripheral phospho-S6 or do not respond to re-feeding? Further, in such experiments, rapamycin treatment is a natural control needed to establish mTORC1 specificity; why didn't the authors use it?

As shown in our earlier publication (Korolchuk *et al.* (2011) *Nat Cell Biol*), translocation of lysosomes to the cell periphery in response to nutrients is a stochastic event which is observed in approximately half of the cell population at any given timepoint. However, what is important for our current manuscript is that, in all cells where lysosomes do translocate to the periphery, we also observe peripheral phospho-S6 staining, thus indicating direct and extremely high correlation between the two events (Fig. S2 E). To go beyond correlation, in addition to ARL8B KD data, which was in our original manuscript, we have now also performed experiments where we forced lysosomes to the periphery. Specifically, we increased the % cells with peripheral lysosomes by over-expressing ARL8B, which we previously showed to promote mTORC1 activity (Korolchuk *et al.* (2011) *Nat Cell Biol*). We found that, consistent with our model, ARL8B increases peripheral phospho-S6 levels and the proportion of cells with peripheral phospho-S6 (Fig. S2 F). Rapamycin controls have been done in the initial phases of the project, however were not included as our starvation/recovery experiments clearly show the response of phospho-S6 to mTORC1 inhibition. However, we agree with the Reviewer that rapamycin controls for phospho-S6 and other readouts for mTORC1 are useful and these are now shown in Fig. 3 and Fig. S3.

4. I am somewhat worried about the results obtained using TORCAR FRET reporter, which appears to be one of the strongest evidence for the authors' conclusions. First, rapamycin should be used after FCS treatment to demonstrate both the reversibility and mTORC1 specificity of the FRET signal.

According to the Reviewer's suggestion, we have now performed TORCAR FRET measurements in the presence of rapamycin. Results of these experiments revealed that rapamycin treatment blocks FCS-mediated increase in mTORC1 activity in focal adhesions (Fig. S3 D).

Loss of FRET is often associated with unequal photobleaching, especially with a relatively low dynamic range of the FRET signal.

We fully agree with the Reviewer that the quantitative FRET analysis is a difficult issue with multiple pitfalls. Therefore, during the control experiments (i.e. TORCAR FRET measurements without FCS), we performed recordings of the fluorescence intensity separately for the donor and acceptor. In Fig. R1 below, you can see a representative graph illustrating that, under our experimental conditions, we did not observe significant and/or unequal bleaching of donor (CFP) and acceptor (YPet) fluorophores. Moreover, experiments in which cells were treated with DMEM without FCS show no difference in FRET ratio over time between focal adhesions and control sites (Fig. S3 D), further confirming physiological origin of FCS-mediated mTORC1 activation in focal adhesions instead of bleaching artefacts.

Figure R1. Representative traces for fluorescence intensity for the donor and acceptor over time in focal adhesion ROIs in cells treated with DMEM without FCS.

Next, GFP and YPet excitation spectra overlap significantly, which could result in weird ratio even after unmixing; the lack of the signal in the "control" (i.e., GFP-free) areas could equally well be interpreted as an unmixing artifact. I'd strongly recommend using far-red FPs for paxillin.

We appreciate the opportunity to clarify this point. To overcome the problem with overlapping GFP and YPet spectra, we combine reference-based spectral imaging using the highly sensitive spectral detector of Zeiss LSM 780 ("lambda mode") with an advanced offline algorithm developed in our lab (Włodarczyk *et al.* (2008) *Biophys J*; Zeug *et al.* (2012) *Biophys J*; Prasad *et al.* (2013) *Methods Cell Biol*). Such methodical combination allows for the rigorous unmixing and quantitative, pixel-based analysis of the FRET signal in cells expressing multiple fluorophores as has been shown in our previous publications (Ulc *et al.* (2019) *Glia*; Prasad *et al.* (2019) *J Cell Sci*; Schill *et al.* (2019) *Commun Biol*; Gorinski *et al.* (2019) *Nat Commun*). We also included in Fig. R2 below representative images showing efficient spectral unmixing in HeLa cells co-expressing cytosolic YPet and GFP-paxillin. Moreover, as mentioned above, our control experiments (i.e. cells treated with DMEM w/o FCS) did not show any changes in FRET ratio over time, further excluding spectral overlapping or bleaching artefacts.

Figure R2. A Representative fluorescence images of cytosolic YPet and GFP-paxillin transfected HeLa cells. Scale bar, upper panel, 20 μm . Scale bar, lower panel, 5 μm . **B** shows the unmixing spectra of GFP and YPet.

Finally, I could not find an example of the FRET ratio image when cells are treated with cell-permeable leucine analog; please add.

In line with the Reviewer's suggestion, we have added representative examples of TORCAR FRET images after treatment of cells with leucine methylester to the revised manuscript (Fig. S3 C).

5. Arl8b KO was shown to significantly delay mTORC1 as well as mTORC2/Akt activity upon re-feeding, with ~30-40% recovered only after 1 hour (Jia and Bonifacino, *Mol Cell* 2019). The authors, however, compare phospho-S6 staining only after 10 min. Does the peripheral mTORC1 pool still contribute to S6 phosphorylation 1 hr after re-feeding? Could PM-localized Akt (and not FAs) play a role in mTORC1 activation at the PM (to remove Tsc2)?

We would like to point out that our manuscript has a focus on the early events in mTORC1 activation in response to growth-promoting stimuli, which is the major area of research in the mTOR field. However, we now performed re-feeding of cells both for 10 and 60 minutes and, in agreement with Jia and Bonifacino (2019) *Mol Cell*, found that ARL8B KD suppresses mTORC1 re-activation at both time points (Fig. S6 E). Furthermore, we found that constitutive targeting of mTORC1 to FAs (see below) rescues the mTORC1 activation defect in ARL8B KD cells at both time points (Fig. S6 E).

Re. the role of PM-localized Akt; although monitoring activation of Akt at the subcellular level is challenging with reagents currently available in the field, we have demonstrated that GFRs and mTORC1 are specifically activated in paxillin-positive FA, so it is highly likely that Akt is also phosphorylated in these regions. The mechanism by which Akt removes TSC from peripheral lysosomes is plausible, however we were unable to detect TSC2 on peripheral lysosomes and therefore could not test this further, but as the mechanisms controlling TSC localisation to the lysosome are still relatively unknown, future studies may shed light on this question.

6. Experiments in Talin1/2 KO cells, adherent/suspension cells and with ROCKi and integrin antagonist are the strongest evidence supporting the authors' hypothesis, but they require proper specificity controls with mTORC1, mTORC2, PI3K and Akt substrates and the corresponding rescue/inhibitor washout experiments wherever possible.

In addition to the previously shown mTORC1 readouts, we now show phospho-Akt S308 and phospho-Akt T473 blots, readouts for PI3K/Akt and mTORC2 activities, respectively (Fig. 5 B). Both readouts are partially suppressed by Talin knockout, consistent with suppression of the entire GFR/PI3K/Akt/mTOR axis upon FA perturbation.

In our experiments with ROCKi and integrin antagonist cilengitide, cells were pre-treated with the drugs for the last 2 hours of the starvation regime. We found that, whether the 10 min re-feeding was performed in the presence or absence (effectively drug washout) of drugs, mTORC1 was equally suppressed (not shown). This indicates to us that either the drugs are irreversible, or 10 min of washout is not sufficient to re-form the FAs. In either case, we do not believe we can provide the informative drug washout data within the feeding protocols used in our study.

7. Finally, I think it is important to better define the authors' working hypothesis. They propose that "peripheral dispersal of mTORC1-positive lysosomes supports mTORC1 activation by bringing it into close proximity to incoming mitogenic signaling cascades and amino acids". Of course this could be the case, but discrimination between the contribution of the peripheral and interior pools of mTORC1 seems to be a non-trivial task, as such

spatially distinct pools could have different catalytic activities, flux of substrates or distinct inactivation kinetics. Unraveling the contribution of these parameters would require careful experimentation.

We have considered this point carefully and believe that our new experiments in the revised manuscript provide support for our original working hypothesis. We do agree, however, that we are only starting to understand the complexity of mTORC1 signalling at the sub-cellular level and there are limitations to our study; we highlighted this in the Discussion section of the revised manuscript.

Reviewer #2 (Comments to the Authors (Required)):

1) The authors provide a nicely illustrated and informative representation of their RPTOR BioID2 dataset that convincingly demonstrates the proximity of mTORC1 with focal adhesions. Interestingly, two of the nine proteins were found significantly enriched in RPTOR/Kindlin/paxillin proximity proteomes are linked predominantly with fibrillar adhesions (tensin-1 and tensin-3) and two are components of the FA proximal structures that link focal adhesion to microtubules (Kank and liprin). These data would imply that mTORC1 proximity is not linked to focal adhesion specifically but possibly engages different types of adhesions in cells. This would also be in line with a previous study showing recruitment of lysosomes and active mTOR to fibrillary adhesions (doi: 10.1016/j.celrep.2014.12.037). Could the authors expand on their data in Figure 2 to also other adhesion types than paxillin-positive adhesions?

We have consulted with Prof. Martin Humphries at the University of Manchester, an expert in cell adhesion, and we are of the understanding that there is no definitive marker for fibrillar adhesions but that localisation within the cell is commonly used to identify such structures and they can, in fact, be paxillin-positive. We also note that tensin-1 and tensin-3 have been described to localise to peripheral focal adhesions (Barber-Perez *et al.* (2020) *J Cell Sci*). We therefore analysed the distribution of paxillin-positive structures and how it correlates with phospho-S6 distribution, i.e. at the cell periphery versus under the centre of the cell. As can be seen in Fig. 3, C and L, and Fig. S3 A, paxillin accumulates in cell periphery regions (corresponding to FAs) and also at lower density under the cell body (which we interpret as being at least in part localised to fibrillar adhesions). Importantly, following refeeding, phospho-S6 (and a readout for newly synthesised proteins, HPG) are enriched in FA regions and also gradually increase towards the middle of the cell. Therefore, we cannot rule out the role of fibrillar adhesions under the cell body in mTORC1 activation, which we stated in the Discussion and cited the study by Rainero *et al.* (2015) *Cell Rep*. However, we believe that our experiments with FA disruption and constitutive targeting of RPTOR to FAs strongly implicate FAs as the key hub for mTORC1 activation.

2) The proximity biotinylation was carried out in osteosarcoma U2OS cells. Why is the validation data of mTOR and LAMP1 recruitment to paxillin positive adhesion with HeLa cells? The authors should consider showing this (and possibly localization to other adhesion types as well) in other cell types to demonstrate the generality/specificity of these data (there is some data from MEFs and MRC5 cells later, but would be nice to have a broader validation here).

The reason for using U2OS cells in our proteomics studies was that these were engineered to carry RPTOR-BioID. To validate localisation of mTORC1 activity, we now performed phospho-S6 stainings in U2OS, and also in Cos7 and HEK293 cell lines. In all of these models, we see a clear phospho-S6 signal in the proximity of FAs upon feeding (Fig. S3 B).

3) Figure 3a. Could the authors somehow demonstrate that the "significant increase in co-localisation between p-S6 and paxillin" is not just a reflection of having low p-S6 in the cell in starved conditions to having it abundantly throughout the cells in fed conditions?

To address this question in a formalised way, we used two approaches. First, we employed analyses similar to those used in our TORCAR experiments, i.e. by quantifying intensity of phospho-S6 in the regions of interest (ROI) defined by peripheral focal adhesions vs neighbouring ROI which are devoid of focal adhesions. New data presented in Fig. 3, D, E and M, clearly indicate that phospho-S6 (and a newly introduced readout for mTORC1, HPG) is significantly higher in the peripheral areas where focal adhesions accumulate compared to adjacent (control) regions. Second, we analysed fluorescence intensity profiles for paxillin and mTORC1 readouts from cell periphery to the nucleus. The histograms shown in Fig. 3, C and L, and Fig. S3 A indicate striking enrichment of phospho-S6 and HPG in the paxillin-positive areas. Based on both approaches we conclude that peripheral distribution of mTORC1 activity is not random but tightly associates with FA areas.

4) Figure 4a. How have the authors validated the specificity/ruled out off-target effects of the ARL8B siRNA?

The effects of our ARL8B siRNA on mTORC1 activity and specificity were validated in our previous paper (Korolchuk *et al.* (2011) *Nat Cell Biol*), using multiple siRNAs (including the ones used in this study) and rescue with ARL8B OE. We made a reference to the original study in the text.

5) Does ARL8B siRNA influence adhesion number or distribution in the cells.

We analysed distribution of paxillin staining in cells with/without ARL8B depletion and did not detect differences (Fig. S6 D), which is consistent with unperturbed growth factor signalling in ARL8B KD cells (Fig. 6 D). We also quantified the number and size of focal adhesions, which are not affected by ARL8B depletion (Fig. R3).

Fig. R3: HeLa cells were treated with scrambled (Scr) or ARL8B siRNA HeLa cells for 96 hours. Cells were stained with antibodies against paxillin and the number (a) and size (b,c) of paxillin-positive FAs was quantified.

6) Starvation in other cell models has been shown to impact on the balance between peripheral focal and adhesions and elongated adhesions localising under the cell body.

The majority of our protocols involve comparison between starved cells and cells re-fed for 10 minutes. Even within this short stimulation period, we see an increase in the intensity of paxillin staining at the cell periphery (FAs), which is concomitant with the reduction of its intensity inside the cell (potentially associated with other types of complexes, such as fibrillar adhesions) (Fig. S3 A). This is consistent with previous reports (Rainero *et al.* (2015) *Cell*

Rep) and also supports the role for FAs as the hubs of nutrient sensing.

7) Figure 4G would be easier to interpret if starved and fed condition were shown like in the other experiments.

There is essentially no phospho-S6 signal in starved cells, which is shown repeatedly in the manuscript; therefore, to avoid duplication and save space, we only show FA inhibitor data for cells stimulated with nutrients (Fig. 5 D). However, starvation conditions have been shown in western blot data (Fig. S5 E) for consistency with other blots.

8) In the discussion the authors speculate that "Activation of mTORC1 at FAs by nutrients may therefore suppress autophagy of FA components". They should mention earlier work describing already that mTOR activity suppressed integrin and ECM protein uptake into spatially localised adhesion proximal lysosomes.

We have referenced the study indicating that inhibition of mTORC1 increases integrin uptake into adjacent lysosomes (Rainero *et al.* (2015) *Cell Rep*) in the Discussion. We also believe that the experiments suggested by the reviewer earlier allowed us to more clearly put our findings in the context of published literature.

Minor points:

9. Figure 2c. What is SDHA? Why was it chosen as control? It might be better to score the number of PLA spots/cell or the number of PLA spots overlapping with Paxillin rather than the % of PLA positive cells.

We have clarified that SDHA is a mitochondrial protein and used here as a negative/unrelated control as mTORC1 has not been previously shown to associate with mitochondria. Quantification of PLA as number of PLA dots/cell shows the same pattern as % of PLA-positive cells (Fig. 2 C).

10. Figure 3c would benefit from adding labelling in the images (to indicate what the green signal is).

Page 7 top, this sentence may need to be edited "although at this point is cannot necessarily be completely ruled out."

The text/figures have been modified.

Reviewer #3 (Comments to the Authors (Required)):

1. Although data analyses were performed at a state-of-the-art level, and the results are clearly represented the manuscript falls short in demonstrating under which physiological conditions the postulated "maximal mTORC1 activation" would be required.

The conditions where cells were fed with a full complement of amino acids and growth factors (i.e. FCS) were considered to induce maximal activation of mTOR. As the term could be misleading, we replaced it with "full activation of mTORC1" throughout the text.

2. In addition, although in the Arl8b KD experiment mTORC1 signaling is compromised, it remains to my opinion unclear if the results obtained can be really assigned to a lack of targeting focal adhesions. It could be equally well possible that Arl8b KD might interfere with mTORC1 signaling by increasing the amount of regulator associating with BORG and thereby sequestering it away from Rags. If one took this into account, the conclusions would be completely different. Therefore, the authors could for instance artificially target mTORC1

to FA (eg. FAT domain on Rheb) and look if this would rescue the signaling defect detected upon Arl8b KD.

As suggested by the Reviewer, we generated constructs expressing RPTOR fused to the focal adhesion targeting (FAT) domain sequence of vinculin. The new data clearly demonstrate that, in agreement with our model and the Reviewer's prediction, forced targeting of mTORC1 to FAs rescues the signalling defect in ARL8B-depleted cells (Fig. 6; and Fig. S6).

3. Importantly, depletion of focal adhesions, either by Talin KD or ROCK inhibitors, has a significant impact on the cytoskeleton. Chemotactic changes and/or alterations in Rac1 activity have previously been shown to regulate both mTORC1 and mTORC2. Therefore, presented experiments should be interpreted with caution since some of the effects observed might be independent of the here described focal adhesion dependent activation of mTORC1.

We acknowledged in the text the potential pleiotropic effect of interference with FA assembly as a limitation of this study. We then went on to rationalise that forced targeting of mTORC1 to FAs would provide another, and potentially more specific, way to investigate the role of FAs in nutrient sensing.

Minor points that should be revisited by the authors:

4. Figure 1: The results from the MS analysis of the BioID2-RPTOR pulldowns is in part surprising. Why is the log2 enrichment for the autophagy related proteins considerably higher than that observed for core mTORC1 subunits?

Autophagy proteins represent an important set of mTORC1 substrates, and since RPTOR plays a significant role in the recruitment of cargo to the mTORC1 complex (e.g. Ahmed *et al.* (2019) *Sci Rep*), it is not surprising that there is a high relative stoichiometry of autophagy proteins encountered by RPTOR.

5. Figure 3: The authors state that "FAs play an important role in the early events of mTORC1 activation in response to nutrients". Unfortunately, there are a few points unclear to this reviewer concerning the Leucine methyl ester stimulation experiment. In the legend from figure 3 it is mentioned that in panel E, "each data point represents a coverslip including several cells" whereas in panel F, "each data point represents a single cell". Also, in the legend from supplementary figure 3 panel A, "each data point represents a coverslip including several cells". Could the authors please explain why the analysis from the Leucine methyl ester stimulation was apparently performed differently from the remaining TORCAR biosensor experiments?

We thank the Reviewer for this comment and apologise for the technical mistake. In the revised manuscript, we unified data presentation in Fig. 3 and Fig. S3 in the way that, in all FRET-related graphs, each data point represents a coverslip.

Further, it would make the authors claim stronger if they could include an independent positive control, e.g. immunoblotting, to detect mTORC1 activation upon Leucine methyl ester stimulation.

Activation of mTORC1 by immunoblotting has been shown previously (Zoncu *et al.* (2011) *Science*; Manifava *et al.* (2016) *eLife*). As mentioned in our response to Reviewer #1, we

have added representative examples of TORCAR FRET images after treatment of cells with leucine methylester to the revised manuscript as a new Fig. S3 C.

6. Supplementary Figure 2: The image quality on subcellular localization of p-4EBP1 after AA refeeding is not convincing. The entire Panel would also profit from colocalization analysis with either LAMP1 or Paxillin, similarly to what is provided in the remaining panels of the same figure.

The quality of the phospho-4EBP1 antibody is significantly less than phospho-S6, and is particularly weak in response to amino acids alone (in the absence of serum). Therefore, instead of pursuing the use of phospho-4EBP1 antibody further, we removed phospho-4EBP1 data and focused on the TORCAR reporter, which is based on 4EBP1 phosphorylation by mTORC1, for more detailed kinetics and localisation analyses (Fig. 3; and Fig. S3).

7. Figure 4b shows quantified integrated densities of pS6 signals in IF experiments. Somewhat surprisingly the decrease of pS6 signal upon ARL8B knock-down is almost the same for peripheral and intracellular pS6. Such data presentation might give an impression to the readers that peripheral (10% of the cell area) vs intracellular regions were not properly defined. Considering an approximately 2.5-fold decrease in total cell number with peripheral p-S6 upon ARL8b depletion (panel c), one would expect a more specific effect for peripheral staining. The same criticism applies to corresponding Supplementary Figures 4b-c.

As indicated in the Methods section, the regions were identified by automated segmentation of the cell images, and therefore quantification is completely unbiased and precisely defined. The data show changes in the *percentage* of phospho-S6 signal, and the two regions (peripheral and intracellular) have been analysed separately. As such, we do not agree that there is a discrepancy in our data. Instead, it strongly suggests (and this is explicitly written in the manuscript) that preventing peripheral distribution of lysosomes not only prevents activation of mTORC1 at the periphery but also intracellularly. As such, we believe our data indicate that peripheral activation is also important for the overall activation of mTORC1 (which is also confirmed by immunoblotting; Fig. 6 D; and Fig. S6 E).

8. Corresponding to panels a-c in Figure 4, panel d shows immunoblotting analyses of total pS6. It is partially misleading because the conditions (aa and aa plus FCS) are different from panels a-c (FCS starvation and FCS stimulation), and more related to Supplementary Figures 4a-d. The exact conditions are not described, but one could assume 18 h FCS starvation (panels a-c) and 1h aa/FCS starvation in panel d.

The old Fig. 4d showed western blot data for conditions that correspond to both, FCS stimulation (shown as IF data in old Fig. 4a-c) and amino acid stimulation (old Supplementary Fig. 4a-c). As such, one blot was used to illustrate changes in mTORC1 activation in response to both stimulation protocols.

In the revised manuscript we have also investigated the effect of ARL8B depletion in cells expressing RPTOR and RPTOR^{FAT}. Since studies in two cell lines increased the amount of data significantly, we decided to focus on feeding these cell lines with complete media and not investigate amino acids alone.

December 31, 2020

RE: JCB Manuscript #202004010R

Dr. Bernadette Carroll
School of Biochemistry,
University Walk,
Bristol BS8 1TD
Biomedical Sciences Building
United Kingdom

Dear Dr. Carroll,

Thank you for submitting your revised manuscript entitled "mTORC1 activity is supported by spatial association with focal adhesions". Thank you for your patience with the re-review process. You will see that the reviewers are largely supportive of publication. We feel that you have done an adequate revision and also addressed the main points from Ref. #3, who was not available to re-review, so we would be happy to publish the manuscript in JCB pending final minor revisions that address the remaining points from Ref. #1 as well as changes necessary to meet our formatting guidelines (see details below). Please let us know if you would like to discuss any of the changes needed for publication.

1) Please clarify the format at resubmission:

- JCB Articles can have up to 10 main and 5 supplementary figures.
- JCB Reports must have a combined "Results and Discussion" section and are limited to 5 main and 3 supplemental figures.

Please be sure to reorganize the supplemental material to meet the limit of the format desired. A figure can use up to one entire page as long as all panels fit on the page. We feel that the paper is a good fit for the Article format and would suggest resubmitting in that format; we would be happy to further discuss as needed.

2) eTOC summary: A 40-word summary that describes the context and significance of the findings for a general readership should be included on the title page. The statement should be written in the present tense and refer to the work in the third person.

- Please include a summary statement on the title page of the resubmission. It should start with "First author name(s) et al..." to match our preferred style.

3) Figure formatting: Scale bars must be present on all microscopy images, including inset magnifications. Please add scale bars to all magnifications on figure 2, 3, 4, 5, 6, S2, S3, S4, S5, S6; to S1A (main image and magnifications)

Molecular weight or nucleic acid size markers must be included on all gel electrophoresis. Please add molecular weight with unit labels on the following panels: 6D, S1B, S6BE

4) Statistical analysis: Error bars on graphic representations of numerical data must be clearly

described in the figure legend. The number of independent data points (n) represented in a graph must be indicated in the legend. Statistical methods should be explained in full in the materials and methods. For figures presenting pooled data the statistical measure should be defined in the figure legends.

5) Materials and methods: Should be comprehensive and not simply reference a previous publication for details on how an experiment was performed. Please provide full descriptions in the text for readers who may not have access to referenced manuscripts.

- For all cell lines, vectors, constructs/cDNAs, etc. - all genetic material: please include database / vendor ID (e.g., Addgene, ATCC, etc.) or if unavailable, please briefly describe their basic genetic features *even if described in other published work or gifted to you by other investigators*
- Please include species and source for all antibodies, including secondary, as well as catalog numbers/vendor identifiers if available.
- Sequences should be provided for all oligos: primers, si/shRNA, gRNAs, etc.
- More information is needed to meet our policy: even if described in other published work, basic procedures must be described. Please double-check the M&M. For instance, more info is needed for FA isolation.
- Microscope image acquisition: The following information must be provided about the acquisition and processing of images:
 - a. Make and model of microscope
 - b. Type, magnification, and numerical aperture of the objective lenses
 - c. Temperature
 - d. imaging medium
 - e. Fluorochromes
 - f. Camera make and model
 - g. Acquisition software
 - h. Any software used for image processing subsequent to data acquisition. Please include details and types of operations involved (e.g., type of deconvolution, 3D reconstitutions, surface or volume rendering, gamma adjustments, etc.).

6) References: There is no limit to the number of references cited in a manuscript. References should be cited parenthetically in the text by author and year of publication. Abbreviate the names of journals according to PubMed.

- Please edit the reference formatting throughout to meet this style.

7) A summary paragraph of all supplemental material should appear at the end of the Materials and methods section.

- Please include one brief descriptive sentence per item.

A. MANUSCRIPT ORGANIZATION AND FORMATTING:

Full guidelines are available on our Instructions for Authors page, <https://jcb.rupress.org/submission-guidelines#revised>. **Submission of a paper that does not conform to JCB guidelines will delay the acceptance of your manuscript.**

B. FINAL FILES:

-- High-resolution figure and video files: See our detailed guidelines for preparing your production-ready images, <https://jcb.rupress.org/fig-vid-guidelines>.

Thank you for this interesting contribution, we look forward to publishing your paper in Journal of Cell Biology.

Sincerely,

Harald Stenmark, PhD
Monitoring Editor, Journal of Cell Biology

Melina Casadio, PhD
Senior Scientific Editor, Journal of Cell Biology

Reviewer #1 (Comments to the Authors (Required)):

The revised version of the manuscript by Rabanal-Ruiz and co-authors provides further evidence for focal adhesions (and/or FA-associated lysosomes) contributing to mTORC1 signaling. The authors have added a few essential controls as well as an interesting experiment proposed by Reviewer #3, linking Arl8-dependent mTORC1 localization to FA with increased phosphorylation of its substrate S6. The authors carefully addressed criticism of all the reviewers and significantly improved the manuscript.

The manuscript provides an important evidence for the existence of spatially distinct mTORC1 pools in a cell, and I do support its publication in JCB. Yet, I would like to point to a few issues; I hope the authors will use these comments to further cinch up their arguments. I thank the authors for the

productive work on the revision.

1. The experiments shown on Fig. 6 and S6 support a critical argument that constitutive localization of mTORC1 to FAs is sufficient for S6 phosphorylation even in the absence of Arl8. On Fig. 6A, the micrographs show paxillin and pS6 staining, whereas the line profiles refer to co-localization with paxillin (and not pS6); this must be just a typo. On Fig. 6B, however, FLAG-RPTORfat staining (red) shows significant intracellular ER-like staining. Could it be that the increased pS6 levels (seen in WBs - 6D, S6E) are simply due to RPTOR overexpression (and increased mTORC1 activity on endomembranes)? Can the authors estimate the fraction of mislocalized FLAG-RPTORfat? Please comment. Also, adding a rapamycin control would further demonstrate the specificity of the peripheral pS6 signal (as in Fig. 3BE).

2. Using data shown on Fig. 5A, the authors argue that in Talin1/2 DKO cells, aa+FCS were unable to activate mTORC1. While WB on 5B definitely demonstrates just that, there is hardly any obvious co-localization of pS6 with LAMP1 in the control cells either. I could also not find any quantitation of the co-localization data.

3. The authors use incorporation of the methionine analog, L-HPG, as a proxy for local mTORC1 activity (Fig. 3J-M). Do the authors assume mTORC1 activates local protein synthesis on FA and/or peripheral lysosomes? The authors themselves refer to this as a hypothetical mechanism in the outlook (p. 13)...

4. The authors postulate FA to be the hubs for the input of GF-, mitogenic and mTORC1 signaling (p. 8) and even state that GF activation is limited to FAs (Discussion, p. 12). In the abstract, the authors claim that "FAs <...> are necessary for both peripheral and intracellular mTORC1 activity". I find this a rather categorical statement and am unsure if this claim could be fully supported by the evidence. Perhaps, one would refer to FAs as contributing to GF downstream (mTORC1) signaling?

The reviewer claims no competing interests.

Reviewer #2 (Comments to the Authors (Required)):

The authors have now addressed all of my concerns and I think including the new scoring/consideration of adhesion type/localisation change as part of this metabolic control has further increased the impact of this interesting study.

Dear Editors,

We are pleased to bring to your attention our revised manuscript, 'mTORC1 activity is supported by spatial association with focal adhesions'.

Please find below a point-by-point response to latest comments from reviewer #1, and please find attached the revised manuscript (submitted with and without text track changes for your convenience).

Please let us know if you require any further information, and thank you for your help throughout the submission and revision process.

Best wishes,
Bernadette and Viktor

Please note as requested, please find our statement re. conflict of interest (which is included in the manuscript): B.V. is a consultant for Karus Therapeutics (Oxford, UK), iOnctura (Geneva, Switzerland) and Venthera (Palo Alto, US) and has received speaker fees from Gilead (Foster City, US). The other authors declare no competing financial interests.

Responses to Reviewer #1:

The revised version of the manuscript by Rabanal-Ruiz and co-authors provides further evidence for focal adhesions (and/or FA-associated lysosomes) contributing to mTORC1 signaling. The authors have added a few essential controls as well as an interesting experiment proposed by Reviewer #3, linking Arl8-dependent mTORC1 localization to FA with increased phosphorylation of its substrate S6. The authors carefully addressed criticism of all the reviewers and significantly improved the manuscript.

The manuscript provides an important evidence for the existence of spatially distinct mTORC1 pools in a cell, and I do support its publication in JCB. Yet, I would like to point to a few issues; I hope the authors will use these comments to further cinch up their arguments. I thank the authors for the productive work on the revision.

1. The experiments shown on Fig. 6 and S6 support a critical argument that constitutive localization of mTORC1 to FAs is sufficient for S6 phosphorylation even in the absence of Arl8. On Fig. 6A, the micrographs show paxillin and pS6 staining, whereas the line profiles refer to co-localization with paxillin (and not pS6); this must be just a typo.

We thank the Reviewer for this comment and apologise for the technical mistake. The labelling on micrographs in Fig. 6A has been corrected.

On Fig. 6B, however, FLAG-RPTORfat staining (red) shows significant intracellular ER-like staining. Could it be that the increased pS6 levels (seen in WBs - 6D, S6E) are simply due to RPTOR overexpression (and increased mTORC1 activity on endomembranes)? Can the authors estimate the fraction of mislocalized FLAG-RPTORfat? Please comment. Also, adding a rapamycin control would further demonstrate the specificity of the peripheral pS6 signal (as in Fig. 3BE).

We used RAPTOR constructs as a tool to artificially target mTORC1 to FA. While increased mTORC1 activity might be due to RPTOR overexpression, we have controlled for this by comparing the RPTOR^{FAT}-expressing cells with those over-expressing wild-type RPTOR. WB analysis would indicate that if anything, the wild-type construct is expressing at higher levels than the mutant therefore we do not think that the phenotype we see is a consequence of over-expression.

We have confirmed, by immunoprecipitation, that mutant RPTOR^{FAT} is able to form a complex with mTORC1, and this is comparable to wild-type RPTOR-mTOR IP so although we cannot rule out some mis-localisation, our data indicate that the mutant is behaving in a similar manner to the wild-type.

We do not believe that the addition of rapamycin to this experiment will significantly improve the data. We have demonstrated the specificity of the p-S6 peripheral signal in previous figures (Fig. 3 A-B and J-M and Fig. S3 A), and we have demonstrated that RPTOR^{FAT} is in complex with mTORC1 therefore we believe the addition of rapamycin here would be repetitive.

2. Using data shown on Fig. 5A, the authors argue that in Talin1/2 DKO cells, aa+FCS were unable to activate mTORC1. While WB on 5B definitely demonstrates just that, there is hardly any obvious co-localization of pS6 with LAMP1 in the control cells either. I could also not find any quantitation of the co-localization data.

We have quantified the colocalisation between Lamp1 and pS6 in control versus TLN1/2 DKO cells and added it as a new Figure 5 B in the manuscript.

3. The authors use incorporation of the methionine analog, L-HPG, as a proxy for local mTORC1 activity (Fig. 3J-M). Do the authors assume mTORC1 activates local protein synthesis on FA and/or peripheral lysosomes? The authors themselves refer to this as a hypothetical mechanism in the outlook (p. 13).

Our model is that mTORC1 on lysosomes is relocated to the cell periphery, thereby coming into close proximity to FA, a region enriched for mitogenic inputs. We propose that mTORC1 promotes local protein synthesis in this region (whether this occurs on lysosomes or via mTORC1-Rheb shuttling between cytoplasm and lysosome, although this latter model is postulated in the field, there is currently little experimental evidence). We do not propose that mTORC1 relocates from lysosomes to FA, and indeed we have no evidence of any direct interaction between mTORC1 and FA proteins. Hopefully this is clear in the text.

4. The authors postulate FA to be the hubs for the input of GF-, mitogenic and mTORC1 signaling (p. 8) and even state that GF activation is limited to FAs (Discussion, p. 12). In the abstract, the authors claim that "FAs <...> are necessary for both peripheral and intracellular mTORC1 activity". I find this a rather categorical statement and am unsure if this claim could be fully supported by the evidence. Perhaps, one would refer to FAs as contributing to GF downstream (mTORC1) signaling?

As suggested, we have modified the abstract to state that FA contribute to mTORC1 activation rather than are necessary.